# Forward-Backward Latent State Inference for Hidden Continuous-Time semi-Markov Chains

**Nicolai Engelmann**
Department of Electrical Engineering
Technische Universitat Darmstadt
64289, Darmstadt
`nicolai.engelmann@tu-darmstadt.de`

**Heinz Köppl**
Department of Electrical Engineering
Department of Biology
Technische Universitat Darmstadt
64289, Darmstadt
`heinz.koeppl@tu-darmstadt.de`

## Abstract

Hidden semi-Markov Models (HSMM's) - while broadly in use - are restricted to a discrete and uniform time grid. They are thus not well suited to explain often irregularly spaced discrete event data from continuous-time phenomena. We show that non-sampling-based latent state inference used in HSMM's can be generalized to latent Continuous-Time semi-Markov Chains (CTSMC's). We formulate integro-differential forward and backward equations adjusted to the observation likelihood and introduce an exact integral equation for the Bayesian posterior marginals and a scalable Viterbi-type algorithm for posterior path estimates. The presented equations can be efficiently solved using well-known numerical methods. As a practical tool, variable-step HSMM's are introduced. We evaluate our approaches in latent state inference scenarios in comparison to classical HSMM's.

## 1 Introduction

Continuous-Time semi-Markov Chains (CTSMC's) comprise the continuous time analogue of (discrete-time) semi-Markov chains. In comparison to Markov chains, semi-Markov chains allow to model non-exponential, arbitrary waiting times between state changes. This more accurately matches such times occurring in many scientific fields [11]. Hidden semi-Markov Models (HSMM's) on finite state spaces have seen wide application and thorough research [22]. However, confronted with measurements from continuous-time systems, their adaptivity is limited and computation might become unnecessarily expensive. Because of that, continuous-time analogues of HSMM's have seen increased research interest lately [17, 1, 6, 7]. However, latent state inference is hard in continuous time, since the usual methods demand a reformulation of the process to obtain the Markov property. Markovianization by state space expansion as used for HSMM's demands propagation of a semi-infinite uncountable state for latent CTSMC's even if the original state space is finite. Thus, besides ad-hoc discretization, two classes of solution methods have emerged. First, a variety of sampling-based solutions has been published, often targeted at even generalized problems [7, 2, 1, 9]. Second, if non-sampling based solutions are sought, state-space expansions based on approximations of waiting time measures are available [16]. This second class, by detour, essentially tries to approximate Kolmogorov's (forward/backward) equations. However, a general framework - not tailored to specific families of waiting time measures - is still lacking. Kolmogorov's equations in integro-differential form have been studied recently [20]. Although initial or terminal states are given, the equations rely on specific boundary conditions to restrict the non-Markovian process to the considered time-window. Fixed conditions hinder the equations' usage in latent state inference which typically spans many such time-windows. To alleviate this restriction, we show how to adjust the memory terms in Kolmogorov's equations to include observations. As a consequence, we provide general forward and a backward algorithms, each encoded in a single inhomogeneous integro-differential

36th Conference on Neural Information Processing Systems (NeurIPS 2022).

equation. We also provide a method for exact Bayesian smoothing which uses the quantities obtained from the forward and backward algorithms. As a corollary, we propose a Viterbi-type approach for maximum-a-posteriori path estimation and sketch a method to implement adaptive step-size HSMM's that keep their homogenous properties. The resulting general framework may incorporate arbitrary numerical solvers or make use of approximations by the mentioned second class of solution methods.

**Related Work.** Continuous-time HMM's and their inference has recently been compiled by [14]). Although tailored to a specific problem, an interesting MAP path estimation method for CT-HSMM's based on stochastic state classes has been published in [5]. Neural ODE's for the progression of memory variables have been investigated in [7]). An application of CT-HSMM's including parameter inference using a discretized waiting time variable [6, 17]. Generalized HSMM's that are lossless continuous discretizations but the transitions are observable has been proposed in [21]. An elaborate forward-filtering-backward-sampling for a HASMM [1]. To our knowledge, we are the first to introduce a "traditional" form of a forward-backward algorithm for a latent CTSMC. Also, while [5] is able to do MAP estimates of trajectories as well, their method is very different to our approach.

**Notation.** To maintain readability, we use the symbol $P$ to denote density and mass functions alike. Which kind is needed should be clear from the context. Nonetheless, we refer to appendix section A for derivations and proofs using a stricter notation style.

## 1.1 Continuous-Time semi-Markov Chains

Given a probability space $(\Omega, \mathfrak{F}, \mathrm{Pr})$ and a measurable space $(\mathcal{X}, \mathfrak{X})$. In this work, we assume $\mathcal{X}$ to be countable. Then, a continuous-time semi-Markov chain (CTSMC) on the state space $\mathcal{X}$ is a stochastic jump process $X : \Omega \times \mathbb{R}_{\geq 0} \to \mathcal{X}$. The general process is described by transition rates $\lambda : \mathcal{X} \times \mathbb{R}_{\geq 0} \times \mathbb{R}_{\geq 0} \times \mathcal{X} \to \mathbb{R}_{\geq 0}$. Conditioned on the current state $x \in \mathcal{X}$ at time $t \in \mathbb{R}_{\geq 0}$ and the time $\tau$ since the last transition at $t - \tau < t$, the rates $\lambda(x, \tau, t; x')$ describe the probability per unit time of transitioning shortly after time $t$ to state $x'$. These transition rates suffice to resolve the waiting times and jump directions for each tuple $(x, \tau)$ at all times $t$. In this work, we restrict ourselves to CTSMC's which are homogeneous and direction-time independent. The former means that $\lambda(x, \tau, t; x') \equiv \lambda(x, \tau; x')$ at all times $t$. The latter additionally implies the factorization $\lambda(x, \tau; x') \equiv m(x' \mid x) \lambda(x, \tau)$ with the exit rate $\lambda(x, \tau) \equiv \sum_{x' \neq x} \lambda(x, \tau; x')$ and the transition probabilities of the embedded Markov chain $m(x' \mid x)$. Satisfying $\sum_{x' \neq x} m(x' \mid x) = 1$, i.e., excluding self-transitions, they form a categorical distribution over next states $x' \neq x$. Under direction-time independence, the waiting time in each state is thus statistically independent of the jump direction.

In this setting, for each state $x$, we have the waiting time measures $F(\tau \mid x) = 1 - \Lambda(\tau \mid x)$ with survival/tail functions $\Lambda(\tau \mid x) = \exp\left(-\int_0^\tau \mathrm{d}\tau' \lambda(x, \tau')\right)$ and their densities $f(\tau \mid x) = \lambda(x, \tau) \Lambda(\tau \mid x)$. For convenience of notation, we also introduce the forward transition operator of the embedded Markov chain $\mathsf{M}$ acting on a test-function $g$ by $\mathsf{M}g(x, t) \equiv \sum_{x' \neq x} m(x \mid x') g(x', t)$ and its backward adjoint $\mathsf{M}^\dagger$, for which $\mathsf{M}^\dagger g(x, t) \equiv \sum_{x' \neq x} m(x' \mid x) g(x', t)$ holds.

Further, a homogeneous continuous-time Markov chain (CTMC) is a homogeneous and direction-time independent CTSMC which has constant rates in each state. As such, the rates $\lambda(x, \tau) \equiv \lambda(x)$ are left dependent only on the current state. Thus, the waiting time measures are strictly exponential and so $F(\tau \mid x) = 1 - \exp(-\lambda(x) \tau)$ and $f(\tau \mid x) = \lambda(x) \exp(-\lambda(x) \tau)$. Though significantly less expressive, inference in CTMC's is by large parts available analytically [18].

## 1.2 Observation Model

In this work, we consider a continuous-time version of a hidden semi-Markov model, where the latent process is a CTSMC $X(t)$ and we are given a set of observations at discrete time instances. This means, for specific $t'$, the state $X(t')$ is observable indirectly by means of a dependent random variable $Y(t')$ taking values in a measurable space $(\mathcal{Y}, \mathfrak{Y})$. We assume to collect a set of observations $y_0, y_1, \ldots, y_K \in \mathcal{Y}$ only at discrete time instances $t_0 < t_1 < \cdots < t_K$ and demand proper likelihood functions $L_k(x \mid y_k)$ be given and known at observation instances $t_k$. Like mentioned above, we will use the notation $P(y_k \mid X(t_k) = x) \equiv L_k(x \mid y_k)$ irrespective of $\mathcal{Y}$ being continuous or discrete. Finally, we will often write events in the form $\mathbf{y}_{[t', t)}$ denoting the joint event obtained by intersecting single observation events, i.e., $\mathbf{y}_{[t', t)} \equiv \bigcap_{t_k \in [t', t)} \{Y(t_k) = y_k\}$.

## 1.3 General Idea

Latent state inference in Markov processes generally relies on the exploitation of their Markov property. It allows to describe the posterior latent state probability as a product of a forward and backward term. While the forward terms account for the probability of past observations at any time instance $t$, the backward terms describe the likelihood of future observations starting from $t$. In the semi-Markov case, such a factorization is generally not possible, i.e. for a state $X(t) = x_t$

$$\underbrace{P\left(x_t \mid \mathbf{y}_{[0,T)}\right)}_{\text{posterior}} \underset{\not\propto}{\not\propto} \underbrace{P\left(x_t \mid \mathbf{y}_{[0,t)}\right)}_{\text{forward}} \times \underbrace{P\left(\mathbf{y}_{[t,T)} \mid x_t\right)}_{\text{backward}}. \tag{1}$$

However, as the name suggests, semi-Markov processes do not always violate the Markov property. In particular, the process is Markov at jump instances [13]. Thus, with respect to state changes, we can again find a factorization similar to (1). To be made precise later, let $\Delta_t$ denote the event of $X(t)$ transitioning to a state $x \in \mathcal{X}$ from a different state $X(t^-) \neq x$ at time instance $t$. Then, in contrast to (1) it holds that

$$\underbrace{P\left(\Delta_t \mid \mathbf{y}_{[0,T)}\right)}_{\text{posterior}} \underset{\propto}{\propto} \underbrace{P\left(\Delta_t \mid \mathbf{y}_{[0,t)}\right)}_{\text{forward}} \times \underbrace{P\left(\mathbf{y}_{[t,T)} \mid \Delta_t\right)}_{\text{backward}}, \tag{2}$$

where $P$ now takes on the form of a density instead of the mass functions in (1). The term $P(\Delta_t \mid \mathbf{y}_{[0,t)})$ in (2) can be calculated under history dependence while the term $P(\mathbf{y}_{[t,T)} \mid \Delta_t)$ is by definition independent of any past trajectory up to $t$ since it is conditioned on the state change at $t$. In the following, we refer to such quantities like those occuring on the right-hand side of (2) as "currents".

## 2 Forward and Backward Currents

Probability currents - in the context of Markov chains - are known from the balance equation [19]. They also occur in the description of other stochastic processes [12], CTSMC's in particular as well [15, 3]. In this section, we establish a general framework of input and output probability and likelihood currents under observations which is needed for the inference methods afterwards.

### 2.1 Input and Output Currents in Continuous-Time semi-Markov Chains

Consider again a CTSMC $\{X(t)\}_{t\geq0}$ on a countable state space $\mathcal{X}$. Let a state $x \in \mathcal{X}$ of the chain and a time instance $t \in \mathbb{R}_{\geq0}$ be given. The following events form the core of the following section. Let $\nabla_h(x,t) \equiv \{X(t) = x, X(t+h) \neq x\}$ denote the joint event of being in two different states $x$ and some $x' \neq x$ at the beginning and end of a time-window of length $h$ past $t$, i.e. the event of leaving state $x$ between $t$ and $t+h$. Let in contrast further $\Delta_h(x,t) \equiv \{X(t) \neq x, X(t+h) = x\}$ denote the joint event of being in state $x' \neq x$ at the beginning of the window and in state $x$ at the end, i.e. entering state $x$ between $t$ and $t+h$. Then, we can quantify the flow of probability into $x$ per unit time with $\phi(x,t) \equiv \lim_{h\to0} \frac{1}{h} P(\Delta_h(x,t))$. We call the quantity $\phi(x,t)$ the input current of state $x$ at $t$. Analogously, we define the output current by $\psi(x,t) \equiv \lim_{h\to0} \frac{1}{h} P(\nabla_h(x,t))$.

In the following, we may omit function arguments if they are the same for all functions. Let the marginal probability of being in state $x$ at $t$ be denoted by $p(x,t) \equiv P(X(t) = x)$. Then, $p$ is related to the input and output currents by

$$\frac{\mathrm{d}}{\mathrm{d}t}p = \phi - \psi = (\mathsf{M} - \mathsf{I})\psi \quad \Leftrightarrow \quad \frac{\mathrm{d}}{\mathrm{d}t}p(x,t) = \sum_{x'}\left[m(x \mid x') - \delta_{x'x}\right]\psi(x',t), \tag{3}$$

where $\mathsf{I}\psi \equiv \psi$ is the identity operator and $\delta$ is the Kronecker delta. This equation is equivalent to the forward equation in an inhomogeneous CTMC and the currents $\phi(x,t)$, $\psi(x,t)$ can directly be derived from it (see appendix section A). The input and output currents are related by the forward transition operator of the embedded Markov chain $\phi = \mathsf{M}\psi$. The following Markov property makes the currents important w.r.t.. (2). Let $\mathbf{x}_{[t,t')}$ denote the trajectory of $X(t)$ in the interval $[t',t)$. Then, $\lim_{h\to0} P\left(\mathbf{x}_{[t,T)} \mid \Delta_h(x,t), \mathbf{x}_{[0,t)}\right) = \lim_{h\to0} P\left(\mathbf{x}_{[t,T)} \mid \Delta_h(x,t)\right)$ for any $x$ and $t$. Thus, the current $\phi(x,t)$ specifies the probability per unit time of the CTSMC having the Markov property at $t$

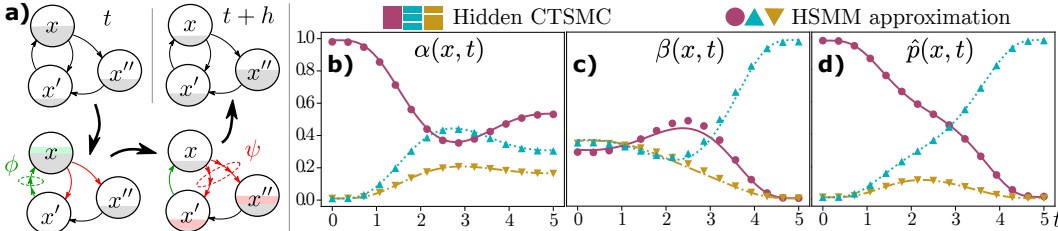

Figure 1: *a)* Illustration of input and output currents to and from state $x$ in a 3-state chain. Like an electric current quantifying the flow of charge across a set of wires, a probability current quantifies the flow of probability across a set of edges in the graph for an infinitesimal duration. Particularly, currents describing probability (and likelihood) flow into and out of single states are important in this work; *b), c), d)* Exemplary calculations of forward $\alpha$ (b), backward $\beta$ (c) and posterior $\hat{p}$ (d) marginals from equations (10) and (12) for only start- and endpoint constraints (Kolmogorov equations) for a three-state chain with Gamma waiting time distributions. The markers show a HSMM approximation with a step-size $h = 10^{-3}$.

next to entering state $x$. On the other hand $\psi(x, t)$ specifies the likelihood of being Markov at $t$ and entering an arbitrary state $x' \neq x$. An illustration of input and output currents is given in Fig. 1.

In this work, we talk about forward or filtered currents specifically if we condition on past observations. For these, we write $\phi_\alpha(x, t) \equiv \lim_{h \to 0} \frac{1}{h} P\left(\Delta_h(x, t) \mid \mathbf{y}_{[0, t)}\right)$ and $\psi_\alpha(x, t) \equiv \lim_{h \to 0} \frac{1}{h} P\left(\nabla_h(x, t) \mid \mathbf{y}_{[0, t)}\right)$. We use the notation $\alpha(x, t) = P\left(X(t) = x \mid \mathbf{y}_{[0, t)}\right)$ for the forward marginals. For the forward currents and marginals (3) holds unaltered. We complement the forward currents with backward currents $\phi_\beta(x, t) \equiv \lim_{h \to 0} P\left(\mathbf{y}_{[t, T)} \mid \Delta_h(x, t)\right)$ and $\psi_\beta(x, t) \equiv \lim_{h \to 0} P\left(\mathbf{y}_{[t, T)} \mid \nabla_h(x, t)\right)$. We also assume a terminal condition to be fixed at $T$. Forward and backward currents share important properties. Let $\beta(x, t) = P\left(\mathbf{y}_{[t, T)} \mid X(t) = x\right)$ be a marginal likelihood complementing $\alpha(x, t)$. Then, we have for $\alpha$, $\beta$ and the both currents

$$\frac{\mathrm{d}}{\mathrm{d}t}\alpha = \phi_\alpha - \psi_\alpha = (\mathsf{M} - \mathsf{I})\psi_\alpha \qquad \Big| \qquad \frac{\mathrm{d}}{\mathrm{d}t}\beta = \psi_\beta - \phi_\beta = \left(\mathsf{I} - \mathsf{M}^\dagger\right)\phi_\beta \tag{4}$$

while the backward currents are related by $\psi_\beta = \mathsf{M}^\dagger \phi_\beta$. Like (3), a resemblance to classical forward and backward equations in a CTMC is not coincidental in (4). We have seen that knowledge of the currents $\phi_\alpha$, $\psi_\alpha$ and $\psi_\beta$, $\phi_\beta$ fully enables calculation of forward and backward terms. In the following, we will outline the necessary equations to determine them.

## 2.2 Obtaining the Forward and Backward Currents through Integral Equations

The input and output currents have an interpretation as a marginalization over paths. This, and their relation to the forward and backward terms (4), enables us to determine how the observations contribute to $\alpha$ and $\beta$ in the history dependent CTSMC. In the following, we will use the shorthand $\mathbf{x}_{[t', t)} = x$ to denote the event of the trajectory $\mathbf{x}_{[t', t)}$ being constant in $X(\tau) = x$, $\forall \tau \in [t', t)$. We begin by resolving a relationship between $p$ and $\phi$ fundamental to CTSMC's. Note, that the probability per unit time of $X(t)$ entering state $x$ at $t - \tau$, $\tau > 0$, and staying in $x$ at least until $t$, can be given by $\Lambda(\tau \mid x)\phi(x, t - \tau) = \lim_{h \to 0} \frac{1}{h} P\left(\mathbf{x}_{[t-\tau, t)} = x, \Delta_h(x, t - \tau)\right)$. To obtain $p(x, t)$, we need to marginalize over all trajectories $\mathbf{x}_{[t-\tau, t)}$ having state $x$ at time $t$. Because of the Markov property at $t - \tau$, we only need to marginalize over all constant trajectories $\mathbf{x}_{[0, t-\tau)} = x$. Thus, we are left to integrate over all $\tau > 0$ to obtain $p(x, t)$. We stop at $\tau = t$ and insert an initial condition from there. Doing this, we obtain the fundamental relationship

$$p = \mathsf{L}\phi + g_p \quad \Leftrightarrow \quad p(x, t) = \int_0^t \mathrm{d}\tau \, \Lambda(t - \tau \mid x)\phi(x, \tau) + g_p(x, t) \tag{5}$$

where $\mathsf{L}$ is a convolutional integral operator defined by $\mathsf{L}\phi(x, t) \equiv \int_0^t \mathrm{d}\tau \, \Lambda(t - \tau \mid x)\phi(x, \tau)$. The inhomogeneity $g_p : \mathcal{X} \times \mathbb{R}_{\geq 0} \to \mathbb{R}_{\geq 0}$ is a known general function and implements the initial condition. Choices for $g_p$ will be discussed in section 4. Instead of asking for $x$ to be kept from $t - \tau$ at least until $t$, we may also demand a transition shortly after $t$ to another state to obtain an

expression for $\psi$. This probability per unit time can then be given by $\lambda\left(x,\,\tau\right)\Lambda\left(\tau\,|\,x\right)\phi\left(x,\,t-\tau\right) = f\left(\tau\right)\phi\left(x,\,t-\tau\right) = \lim_{h\to 0}\frac{1}{h}P\left(\nabla_h\left(x,\,t-\tau\right),\,\mathbf{x}_{[t-\tau,\,t)} = x,\,\Delta_h\left(x,\,t-\tau\right)\right)$. As the result of a marginalization over trajectories similar to (5), i.e. integration over $\tau > 0$, we then obtain

$$\psi = \mathsf{F}\phi + g_\phi \tag{6}$$

with the integral operator $\mathsf{F}\phi\left(x,\,t\right) \equiv \int_0^t \mathrm{d}\tau\, f\left(t-\tau\,|\,x\right)\phi\left(x,\,\tau\right)$. Again, $g_\phi$ implements the initial condition. Note, that we can understand this as an expectation $\psi\left(x,\,t\right) = \mathsf{E}\left[\phi\left(x,\,t-\,\cdot\,\right)\right]$ over the waiting time in state $x$. Differentiating (5) w.r.t. time $t$ and comparing the result with (3) and (6), we get the relation of the inhomogeneities $\frac{\mathrm{d}}{\mathrm{d}t}g_p + g_\phi = 0$. Remembering that $\phi = \mathsf{M}\psi$, we can obtain the two identities $\psi = \mathsf{FM}\psi + \mathsf{M}g_\psi$ and $\phi = \mathsf{MF}\phi + g_\phi$ where $g_\phi = \mathsf{M}g_\psi$.

In the following, we account for observations and then close with the backward currents. Let $\upsilon\left(x,\,t',\,t\right) \equiv P\left(\mathbf{y}_{[t',\,t)}\,|\,\mathbf{x}_{[t',\,t)} = x\right)/P\left(\mathbf{y}_{[t',\,t)}\,|\,\mathbf{y}_{[0,\,t')}\right)$ be the normalized joint observation likelihood of the constant latent trajectory $\mathbf{x}_{[t',\,t)} = x$. The normalizer $P\left(\mathbf{y}_{[t',\,t)}\,|\,\mathbf{y}_{[0,\,t')}\right)$ is the same as implicitly used when normalizing in a CTMC filtering update. The $\upsilon\left(x,\,t',\,t\right)$ is then a $t$-family of piecewise constant functions with discontinuities in the $t'$-direction (or vice-versa). For convenience, we define the modified integral operator $\mathsf{F}_\alpha\phi\left(x,\,t\right) \equiv \int_0^t \mathrm{d}\tau\, f_\upsilon\left(\tau,\,t\,|\,x\right)\phi\left(x,\,\tau\right)$, where $f_\upsilon\left(\tau,\,t\,|\,x\right) \equiv \upsilon\left(x,\,\tau,\,t\right)f\left(t-\tau\,|\,x\right)$. We also define $\mathsf{L}_\alpha\phi\left(x,\,t\right)$ for $\Lambda_\upsilon\left(\tau,\,t\,|\,x\right) \equiv \upsilon\left(x,\,\tau,\,t\right)\Lambda\left(t-\tau\,|\,x\right)$ analoguously. Let further $\upsilon_0\left(x,\,t\right) \equiv \upsilon\left(x,\,0,\,t\right)$ be a shorthand for $\upsilon$ in the interval $[0,\,t]$. Then, we can generalize (5) and (6) to obtain

$$\alpha = \mathsf{L}_\alpha\phi_\alpha + \upsilon_0 g_p \quad\text{and}\quad \psi_\alpha = \mathsf{F}_\alpha\phi + \upsilon_0 g_\phi. \tag{7}$$

From (7) immediately follow the identities $\psi_\alpha = \mathsf{F}_\alpha\mathsf{M}\psi_\alpha + \upsilon_0\mathsf{M}g_\psi$ and $\phi_\alpha = \mathsf{MF}_\alpha\phi_\alpha + \upsilon_0 g_\phi$. The backward currents obey relations analogous to (7). In the following, we will perform the similar construction. Consider the product $f\left(\tau\,|\,x\right)\psi_\beta\left(x,\,t+\tau\right) = \lim_{h\to 0}P\left(\mathbf{y}_{[t,\,T)},\,\nabla_h\left(x,\,t+\tau\right),\,\mathbf{x}_{[t,\,t+\tau)} = x\,|\,\Delta_h\left(x,\,t\right)\right)$. It gives the likelihood of keeping state $x$ in the interval $[t,\,t+\tau)$, then leaving, and along the path encountering any future observations, given an entry in state $x$ at $t$. Thus, $\phi_\beta\left(x,\,t\right)$ can be obtained again by marginalization over future trajectories. The marginal likelihood $\beta\left(x,\,t\right)$ can also be shown to maintain an analogous relationship with $\Lambda\left(\tau\,|\,x\right)\psi_\beta\left(x,\,t+\tau\right)$. This is obtained by a two-side marginalization along state entry $\Delta_h\left(x,\,t-\tau\right)$ and exit $\nabla_h\left(x,\,t+\tau'\right)$ while implying no information on the time of entry $t-\tau$. We refer to appendix section A for the derivation, since it takes more steps. In the reverse direction as we did for the forward case (7), we similarly normalize $\beta$ and the backwards currents $\psi_\beta,\,\phi_\beta$ by $P\left(\mathbf{y}_{[t,\,t')}\,|\,\mathbf{y}_{[0,\,t)}\right)$. This requires a full forward pass to be performed before the backward calculations. Define the integral operators $\mathsf{F}_\beta\psi\left(x,\,t\right) \equiv \int_t^T \mathrm{d}\tau\, f_\upsilon\left(t,\,\tau\,|\,x\right)\psi\left(x,\,\tau\right)$ and $\mathsf{L}_\beta\psi\left(x,\,t\right)$ for $\Lambda_\upsilon\left(t,\,\tau\,|\,x\right)$ analogously. Let further again $\upsilon_T\left(x,\,t\right) \equiv \upsilon\left(x,\,t,\,T\right)$ be a shorthand for $\upsilon$ in the interval $[t,\,T]$. As a conclusion, we obtain the relations

$$\beta = \mathsf{L}_\beta\psi_\beta + \upsilon_T\gamma_p \quad\text{and}\quad \phi_\beta = \mathsf{F}_\beta\psi_\beta + \upsilon_T\gamma_\psi, \tag{8}$$

where the inhomogeneities $\gamma_p$ and $\gamma_\psi$ implement the terminal condition. Note, that we write $\gamma$ to distinguish from the $g$ for the initial conditions. Again by differentiation and consultation of (4), we can find $\frac{\mathrm{d}}{\mathrm{d}t}\gamma_p + \gamma_\phi = 0$. Like before, the recurrence relations $\phi_\beta = \mathsf{M}^\dagger\mathsf{F}_\beta\phi_\beta + \upsilon_T\gamma_\psi$ and $\phi_\beta = \mathsf{F}_\beta\mathsf{M}^\dagger\phi_\beta + \upsilon_T\mathsf{M}^\dagger\gamma_\phi$ follow. Also, $\gamma_\psi = \mathsf{M}^\dagger\gamma_\phi$. Note, that if there are no observations and we only keep the terminal condition, a particular case of both equations in (8) similar to (5) and (6) is obtained.

## 3 Integro-Differential Forward and Backward Equations

We first briefly sketch the derivation of Kolmogorov's forward equation. Our goal is a direct relation between $\frac{\mathrm{d}}{\mathrm{d}t}p$ and $p$. We start out with (3) relating $\frac{\mathrm{d}}{\mathrm{d}t}p$ and $\psi$. We then have (6) expressing $\psi$ in terms of $\phi$ and we also have (5) expressing $p$ in terms of $\phi$. Convolutional equations can be conveniently treated using the Laplace transform. We use it on (5) to express $\phi$ by a convolution of $p$, then plug the result into (6) to express $\psi$ in terms of $p$. The latter result is then inserted into (3) to express $\frac{\mathrm{d}}{\mathrm{d}t}p$ in terms of $p$. The result in the time domain is then

$$\frac{\mathrm{d}}{\mathrm{d}t}p = \left(\mathsf{M} - \mathsf{I}\right)\mathsf{K}p + g \;\Leftrightarrow\; \frac{\mathrm{d}}{\mathrm{d}t}p\left(x,\,t\right) = \sum_{x'}\int_0^t \mathrm{d}\tau\,\left[m\left(x|x'\right) - \delta_{x'x}\right]\kappa\left(x',\,t-\tau\right)p\left(x',\,\tau\right) + g\left(x,\,t\right)$$

$$\tag{9}$$

with the integral operator $\mathsf{K}p\,(x,\,t) \equiv \int_0^t \mathrm{d}\tau \, \kappa\,(x,\,t-\tau)\,p\,(x,\,\tau)$ and the Kronecker delta $\delta$. The memory kernel $\kappa$ is then given by $\mathcal{L}\,\{\kappa\}\,(x,\,s) \equiv \mathcal{L}\,\{f\}\,(s\,|\,x)\,/\,\mathcal{L}\,\{\Lambda\}\,(s\,|\,x)$ with the symbol $\mathcal{L}$ denoting the Laplace transform with the Laplace variable $s$. The inhomogeneity $g$ will be discussed at the end of the section.

In the following, we implement observations and introduce the backward case. Equation (9) is not simply understood as a path marginalization like (5) or (6). However, the path-wise contribution of the observation likelihood carries over from (7) and (8) to equation (9) if the scaled likelihood function $\upsilon\,(x,\,t',\,t)$ is piece-wise constant. The analogue can be shown for the backward case. The rationale is to subdivide the time-domain into intervals where $\upsilon\,(x,\,t',\,t)$ is constant under variation of either $t$ in the forward case or $t'$ in the backward case. The two-argument functions $\bar{\upsilon}\,(x,\,\tau) \equiv \upsilon\,(x,\,\tau,\,t)$ (forward) and $\underline{\upsilon}\,(x,\,\tau) \equiv \upsilon\,(x,\,t',\,\tau)$ (backward) are then simple stepped functions under application of $\mathcal{L}$. The following derivations are executed in detail in Appendix section A.

To obtain the forward equation, we repeat the steps to obtain (9) but instead use our extended equations (7) for each time interval. For the backward equation, we do the following steps. With (4) and the two equations in (8) we have a similar toolset to the forward case. Thus, we apply the Laplace transform to the left equation of (8). We then insert $\psi_\beta$ expressed through $\beta$ into the right equation of (8) which gets us $\phi_\beta$ in terms of $\beta$. Inserting this expression into (4) yields the backward equation valid for the considered time-interval. Joining the obtained sequences of equations back together in both cases yields the integro-differential forward and backward equations

$$\frac{\mathrm{d}}{\mathrm{d}t}\alpha = (\mathsf{M} - \mathsf{I})\,\mathsf{K}_\alpha\alpha + \upsilon_0 g \qquad \text{and} \qquad \frac{\mathrm{d}}{\mathrm{d}t}\beta = \left(\mathsf{I} - \mathsf{M}^\dagger\right)\mathsf{K}_\beta\beta + \upsilon_T \gamma \qquad (10)$$

with the modified integral operators $\mathsf{K}_\alpha\alpha\,(x,\,t) \equiv \int_0^t \mathrm{d}\tau \, \kappa_\upsilon\,(\tau,\,t,\,x)\,p\,(x,\,\tau)$ and $\mathsf{K}_\beta\beta\,(x,\,t) \equiv \int_t^T \mathrm{d}\tau \, \kappa_\upsilon\,(x,\,t,\,\tau)\,\beta\,(x,\,\tau)$ where $\kappa_\upsilon\,(x,\,t',\,t) \equiv \upsilon\,(x,\,t',\,t)\,\kappa\,(x,\,t-t')$. Both inhomogeneities $g$ and $\gamma$ can be built from the inhomogeneities $g_p$, $g_\phi$ and $\gamma_p$, $\gamma_\psi$ from section 2.2. They satisfy the equations $g \equiv (\mathsf{M} - \mathsf{I})\,(g_\phi - \mathsf{K}_\alpha g_p)$ and $\gamma \equiv \left(\mathsf{I} - \mathsf{M}^\dagger\right)(\mathsf{K}_\beta\gamma_p - \gamma_\psi)$. We will consider these in the next section. Finally, note that $\alpha$ and $\beta$ can not simply be multiplied to obtain the full Bayesian posterior $\hat{p}$ w.r.t. the observations. How $\hat{p}$ can be obtained will be discussed in the next section.

## 4 Latent State Inference

There are two possibilities to realize the forward and backward algorithms. We exemplify this on the forward algorithm in the following. In popular HSMM literature [22], the discrete-time equivalent of the products $\phi_\alpha$ and/or $\psi_\alpha$ are fully calculated and $\alpha$ is derived from them using (3). This general procedure poses the computationally most efficient overall solution under two circumstances. First, if we only aim to calculate the full Bayesian posterior $\hat{p}\,(x,\,t) = P\left(X\,(t) = x\,|\,\mathbf{y}_{[0,\,T)}\right)$ for all $t$ we don't need $\alpha$ itself but only $\phi_\alpha$ and/or $\psi_\alpha$. The currents $\phi_\alpha$, $\psi_\alpha$ are obtained from (7) and (10) with comparable effort (c.f. section 4.2). Second, if the classical memory kernel $\mathsf{K}$ and thus $\mathsf{K}_\alpha$ is hard to evaluate (e.g. by numerical inverse Laplace transform), an evaluation of $\mathsf{F}$ and thus $\mathsf{F}_\alpha$ might be computationally preferable, even in the case where we aim to obtain $\alpha$ from (4) or (7) afterwards. In this work, we chose to evaluate $\alpha$ and $\beta$ directly from (10) using a finite-element approach. Thus, we introduce a time-mesh $t_0 < t_1 < \cdots < t_K$, where we set each $t_k$ exactly at the times of observation.

**Initial and Terminal Conditions.** Until now, we assumed the boundary conditions $g$ and $\gamma$ to be known but didn't specify them. Compared to CTMC's, the boundary conditions are not single states but semi-infinite trajectories. In HSMM literature [22], there are two popular boundary conditions. The first assumes, a transition occures both before $t_0$ and after $t_K$. The second assumes a steady state before $t_0$ and an uninformed progression after $t_K$. The latter corresponds to the case that is classically expressed by a vector of ones in HMM's when the backward term is initialized. More on boundary conditions and their exact expressions are found in appendix section A.

### 4.1 Unidirectional Forward and Backward Algorithms

We formulate the general forward and backward algorithms in appendix section C. The interpolants $\alpha_k$ and $\beta_k$ valid in the $k$-th time interval $t \in [t_k,\,t_{k+1})$ obey the following integral equations

$$\frac{\mathrm{d}}{\mathrm{d}t}\alpha_k = (\mathsf{M} - \mathsf{I})\,\mathsf{K}_{[t_k,\,t)}\alpha_k + g_k \qquad \Big| \qquad \frac{\mathrm{d}}{\mathrm{d}t}\beta_k = \left(\mathsf{I} - \mathsf{M}^\dagger\right)\mathsf{K}_{(t,\,t_{k+1}]}\beta_k + \gamma_k \qquad (11)$$

where we introduce the truncated integral operators $\mathsf{K}_{[a,\,b)}\alpha_k \equiv \int_a^b \mathrm{d}\tau \, \kappa\,(t-\tau,\,x)\,\alpha_k\,(x,\,\tau)$ and $\mathsf{K}_{(a,\,b]}\beta_k \equiv \int_a^b \mathrm{d}\tau \, \kappa\,(\tau-t,\,x)\,\beta_k\,(x,\,\tau)$. The inhomogeneities $g_k$ and $\gamma_k$ are built during runtime. We build $g_k$ from the overall boundary condition $g_0$, the previous interpolants $\alpha_{k'}$, $k' < k$, and the scaled likelihood function $\upsilon$. The backward $\gamma_k$ is built analogously with $\gamma_K$ and $\beta_{k'}$, $k' > k$. Note, that the data likelihood $P\left(\mathbf{y}_{[0,\,T)}\right)$ usable to infer model parameters is obtained in the forward pass.

## 4.2 Forward-Backward Algorithm - Posterior Marginals

Obtaining the posterior marginals is possible on the level of currents. Since they mark times when the CTSMC has the Markov property, we can build the product of the two normalized currents $\psi_\alpha(x,\,t) = \lim_{h\to 0} \frac{1}{h} P(\nabla_h(x,\,t)\,|\,\mathbf{y}_{[0,\,t)})$ and $\psi_\beta(x,\,t) = \lim_{h\to 0} P(\mathbf{y}_{[t,\,T)}\,|\,\nabla_h(x,\,t))/P(\mathbf{y}_{[t,\,T)}\,|\,\mathbf{y}_{[0,\,t)})$ to obtain the posterior current $\hat{\psi}(x,\,t) = \lim_{h\to 0} \frac{1}{h} P(\nabla_h(x,\,t)\,|\,\mathbf{y}_{[0,\,T)})$. We can then use $\hat{\psi}$ in (3) to obtain the derivative $\frac{\mathrm{d}}{\mathrm{d}t}\hat{p}$. We discuss this direct approach in appendix section A. Although the former is preferable from a runtime perspective, an more classical approach popular in discrete-time HSMM literature [22] uses a two-sided path marginalization which allows usage of interpolation-based numerical methods. Thus, we calculate the posterior marginals from $\phi_\alpha$ and $\psi_\beta$ via

$$\hat{p} \propto \Pi_f\left(\phi_\alpha,\,\psi_\beta\right) + \mathsf{G}\psi_\beta + \Gamma\phi_\alpha \tag{12}$$

with the integral operator $\Pi_f\left(\phi_\alpha,\,\psi_\beta\right) \equiv \int_0^t \mathrm{d}\tau \int_t^T \mathrm{d}\tau' \, \phi_\alpha\,(x,\,t-\tau)\,f_\upsilon\,(\tau,\,\tau'\,|\,x)\,\psi_\beta\,(x,\,t+\tau')$ and the boundary integral operators $\mathsf{G}\psi_\beta \equiv \int_0^{T-t} \mathrm{d}\tau \, g\,(x,\,t+\tau')\,\upsilon_0\,(x,\,t+\tau')\,\psi_\beta\,(x,\,t+\tau')$ and $\Gamma\phi_\alpha \equiv \int_0^t \mathrm{d}\tau \int_t^T \mathrm{d}\tau' \, \phi_\alpha\,(x,\,t-\tau)\,\upsilon_T\,(x,\,t-\tau)\,\gamma\,(x,\,t-\tau)$. A written out version of (12) is available in Appendix section A. We need to add that the currents $\phi_\alpha$ and $\psi_\beta$ are obtained as by-products without additional computation from the unidirectional forward and backward algorithms introduced in section 4.1. An algorithmic description and an explanation how $\phi_\alpha$ and $\psi_\beta$ are obtained as by-products can be found in Appendix section C.

## 4.3 Viterbi-type Algorithm - Latent Trajectory Point Estimation

Besides latent state marginals, point estimates of whole posterior paths are as well important in further inference tasks. Originally stemming from HMM's, the Viterbi algorithm which calculates MAP path estimates has been successfully adopted to HSMM's [22]. However, like in hidden CTMC's, the unknown chain length, i.e. the number of state changes until the terminal state is reached, makes the algorithm hard to translate to continuous time. One is therefor often content with knowing the MAP state only at observation times [14]. The latter can in principle be obtained using our forward algorithm in section 4.1. Nonetheless, for a finite number of jumps, the framework of currents allows the formulation of a scalable full-path algorithm. A max-product-type reformulation of (7) allows an introduction of a max-input current

$$\phi_{\max}\,(x,\,t;\,n) \equiv \sup_{\mathbf{x}_{[0,\,t)}} \lim_{h\to 0} \frac{1}{h} P\left(\Delta_h\,(x,\,t),\,\mathbf{x}_{[0,\,t)}\,|\,\mathbf{y}_{[0,\,t)},\,n\right)$$

which - bluntly speaking - substitutes the marginalization over all $\mathbf{x}_{[0,\,t)}$ in $\phi_\alpha$ with a supremum. To keep track of the jump count, we equip this current with an additional parameter $n$. We also introduce a maximizing transition operator $\mathsf{M}_{\max}$ for the embedded Markov chain $\mathsf{M}_{\max}g\,(x,\,t) \equiv \max_{x'\neq x} m\,(x\,|\,x')\,g\,(x',\,t)$. The transition operator $\mathsf{M}_{\max}$ resembles $\mathsf{M}$ where the sum over the set $\mathcal{X} \setminus \{x\}$ is exchanged by a maximization. The constituting recursion for $\phi_{\max}\,(x,\,t;\,n)$ is then given by

$$\phi_{\max}\,(x,\,t;\,n+1) = \sup_{x'\neq x,\,\tau\in\mathbb{R}_+} m\,(x\,|\,x')\,f\,(t-\tau\,|\,x')\,\upsilon\,(x',\,t-\tau,\,t)\,\phi_{\max}\,(x',\,\tau;\,n) \tag{13}$$

which calculates the max-currents $\phi_{\max}\,(x,\,t;\,n+1)$ associated with the incremented chain length $n+1$. A detailed description is found in Appendix section A. The MAP terminal state $x$ is related to the currents $\phi_{\max}\,(x,\,t;\,n)$ by

$$\sup_{\mathbf{x}_{[0,\,T)}} P\left(\mathbf{x}_{[0,\,T)}\,|\,\mathbf{y}_{[0,\,T)}\right) = \sup_{x,\,\tau,\,n} \Lambda\,(t-\tau\,|\,x)\,\upsilon\,(x,\,t-\tau,\,t)\,\phi_{\max}\,(x,\,\tau;\,n)\,P\left(n\,|\,\mathbf{y}_{[0,\,T)}\right)$$

Let's first assume we found a maximizing tuple $(x^*,\,\tau^*,\,n^*)$. Then, if $n^* > 0$, we can insert $\phi_{\max}\,(x^*,\,\tau^*;\,n^*)$ in (13) and proceed to trace back states and transition instances until the jump

counter reaches zero. However, to find the full tuple $(x^*, \tau^*, n^*)$, we need knowledge of the posterior chain length $P(n \,|\, \mathbf{y}_{[0, T)})$ for $n$ up to infinity, which is intractable. Nonetheless, $P(n \,|\, \mathbf{y}_{[0, T)})$ can be obtained recursively and knowledge of the accumulated probability mass until a specific $n'$ lets us bound the remaining mass for all $n > n'$ letting us know if we passed all high-probability regions. For the recursion, two more sets of input currents $\phi_n (x, t) \equiv \lim_{h \to 0} \frac{1}{h} P\left(\Delta_h(x, t), n \,|\, \mathbf{y}_{[0, t)}\right)$ and marginals $p_n (x, T) \equiv P\left(X(T) = x, n \,|\, \mathbf{y}_{[0, t)}\right)$ are introduced. These are then updated by

$$\phi_{n+1} = \mathsf{MF}_\alpha \phi_n \qquad \text{and} \qquad p_n (x, T) = \int_0^T \mathrm{d}\tau\, \Lambda\left(T - \tau \,|\, x\right) \upsilon\left(x, \tau, T\right) \phi_n\left(x, \tau\right) \qquad (14)$$

The currents are initialized by $\phi_0 (x, t) = \upsilon(x, 0, t)\, g_\phi(x, t)$. Note that the left equation in (14) does not converge to the fixed-point $\phi_\alpha$ but instead will vanish for $n \to \infty$. The posterior chain length $P\left(n \,|\, \mathbf{y}_{[0, t)}\right) = \sum_x p_n (x, T)$ can then be obtained by marginalization. Again, we refer to appendix section A. While we still need to evaluate all equations for arbitrarily large $n$ for an exact result, in most practical scenarios, the most likely number of transitions $n^*$ might be feasibly obtained by bounding. We stress that the bottleneck of our algorithm is the number of transitions, not their temporal closeness.

# 5 Adaptive Step-Size HSMM

One of the major advantages of continuous-time formulations is the availability of adaptive discretization [8, 4]. Although many continuous problems eventually need to be solved numerically on a discrete grid, adaptive step-size methods can bound the discretization error and allow a reduction of the amount of function evaluations. While much more elaborate techniques with tight error bounds are available for integral equations [10], we briefly sketch a simple adaptive discretization strategy to close with a practical perspective on the contributions in this work.

This strategy consists of maintaining a time-dependent set of samples of approximate currents and their timestamps $\Phi_\alpha(t) \equiv \{(\phi_{\alpha, k}, t_k) \,|\, t_K - t_0 < \Delta_{\max}, t_0 < \cdots < t_K, k = 0, \ldots, K\}$. An analogue set is built for $\psi_\beta$. Then, by inserting polynomial interpolants (e.g. affine combinations of adjacent $(\phi_{\alpha, k}, t_k)$ and $(\phi_{\alpha, k+1}, t_{k+1})$), we can reformulate (7) so that it only contains the $\phi_{\alpha, k}$, $t_k$ and definite integrals of the $f(\cdot \,|\, x)$. For these integrals, look-up tables are easily generated for speedup. The resulting linear system can then be solved to calculate the next step $\phi_\alpha^{(1)}(x, t_K + h)$. As a commonly adapted update rule for step size $h$, we use half step checks $\phi_\alpha^{(0.5)}(x, t_K + h)$, where we do two steps using $\frac{h}{2}$ as step size and then calculate $E \equiv |\phi_\alpha^{(1)}(x, t_K + h) - \phi_\alpha^{(0.5)}(x, t_K + h)|$. The integration error $E$ is then used in the update $h \to \min\{\max\{\min\{hs, h_{\max}\}, h_{\min}\}, t' - t_K - h\}$ with the scaling $s = \max\{\min\{\frac{\mathrm{tol}}{E}, s_{\max}\}, s_{\min}\}$ for some pre-specified $h_{\min}, h_{\max}, s_{\min}, s_{\max}$ and the next observation time $t'$.

The resulting HSMM is then only dependent on time-differences, i.e. still homogeneous, but adapts its step-size. In appendix section C, we provide the resulting update equation if affine combinations are used as interpolants between the points $(\phi_{\alpha, k}, t_k)$ and an implicit Euler step rule is chosen.

# 6 Evaluation

## 6.1 Calculation of Integral Equations

Our algorithms demand the solution of integral equations of the second kind. For each time interval $[t, t')$ between observations, we used a backward Euler method to solve the possibly stiff equations in (7), (8) or (10) with a fixed number of steps calculated by $N \equiv \lfloor \frac{t' - t}{h} \rfloor + 1$. Then, all steps are of equal length $h$ except the one at the end of the interval which scales dependent on $t'$. The integration involved in each step is performed by interpolation between the steps, where the trapezoidal rule has been used. This method is outlined in detail in Appendix section B. An exception is the evaluation of the adaptive step-size HSMM. As described in section 5, there the step size is initialized by a smallest value $10^{-4}$ and then the adaptive procedure generates any further grid positions. The integration at each step is also more precise. The details are outlined in Appendix B. For forward and backward passes, each interval is alculated consecutively while the inhomogeneities $g$ and $\gamma$ are updated upon entering each new interval as outline in the algorithm section in Appendix C. The resulting continuous

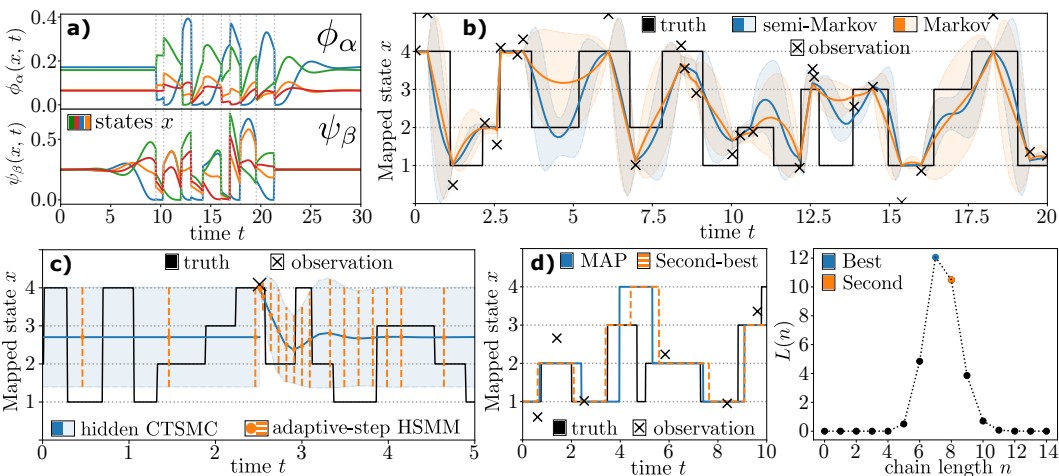

Figure 2: *a)* Normalized forward and backward currents ($\phi_\alpha$, $\psi_\beta$) of a latent CTSMC in presence of observations (vertical gray dotted lines). The areas outside the actual trajectory show the initial and terminal conditions (extending to infinity); *b)* Mean and variance progression of the posterior latent state marginals $\hat{p}$ inferred from the shown trajectory as the original CTSMC (blue) the optimal steady-state CTMC approximation (orange); *c)* Time grid generated by an adaptive step-size HSMM in a forward pass as proposed in section 5 (orange markers). The continuous blue line was generated by a discrete HSMM with step size $h = 10^{-3}$; *d)* MAP latent trajectory estimate for a sample trajectory from a four-state hidden CTSMC. Left: estimated trajectory (blue), second-best candidate (orange). Right: the chain length posterior $P\left(n \mid \mathbf{y}_{[0, t)}\right)$ for different chain lengths $n$.

functions $\alpha$, $\beta$, $\hat{p}$ and the currents are then generated for each interval using cubic spline interpolation on the calculated steps.

## 6.2   Simulations

We calculated several randomly generated hidden CTSMC's. The number of states has been given, but waiting time distributions and transition probabilities were drawn randomly. For each state, we drew a randomly parametrized waiting time distribution from a specified family of distributions (we allowed Gamma and Weibull). The random parameters, were drawn from Gamma distributions with per-parameter calibrated hyperparameters. The transition probabilities of the embedded Markov chains were sampled as sets of random categorical distributions while prohibiting self-transitions. The categorical distributions were sampled by first drawing a vector uniformly from the $|\mathcal{X}|$-hypercube and then projecting it to the $(|\mathcal{X}| - 1)$-simplex via normalization. The observation times were drawn from a point process with Gamma inter-arrival times with fixed shape and rate hyperparameters. The observation values were noisy observations of a function $b : \mathcal{X} \to \mathbb{R}$ mapping the latent states to fixed values in a real observation space $\mathcal{Y} \equiv \mathbb{R}$ with the Borel $\sigma$-algebra $\mathfrak{Y} \equiv \mathfrak{B}$. Only single points were allowed to be drawn, thus requiring density evaluations in the likelihood function $\upsilon$. Therefore, $Y(t) \sim \mathcal{N}(b(X(t)), d)$ with the standard deviation $d$ as a hyperparameter.

The calculated $\alpha$, $\beta$, $\hat{p}$ were compared to those calculated by a HSMM. With a step-size of $h_{\text{HSMM}} = 10^{-4}$ and $h_{\text{CTSMC}} = 10^{-3}$ (backward Euler step-size), no difference was visible anymore on any calculated trajectory. In Fig. 2a, we show an exemplary progression of the forward input $\phi_\alpha$ and backward output currents $\psi_\beta$. In Fig. 2b the smoothed marginals $\hat{p}$ of a four-state CTSMC are compared to those obtained from its steady-state CTMC approximation [13]. For both, smoothed mean and variance of $b(X(t))$ are shown. The full trajectory of $b(X(t))$ by the original CTSMC which generated the observations is shown as a ground truth. We built the adaptive step-size HSMM using our proposed method and ran it over the sampled trajectories as well. While quickly enlarging its step-size, it remained within the margins of a total error of $10^{-3}$ compared to a fine-grained HSMM, although this is not guaranteed by the method. An example run is shown in Fig. 2c. Finally, we tested our Viterbi MAP estimator on the trajectories. An example run is shown in 2d. The posterior chain length marginal $P\left(n \mid \mathbf{y}_{[0, t)}\right)$ had a comparably sharp peak in all trajectories. We assume, this came from the observations greatly narrowing down the amount of candidate transition counts.

# 7 Conclusion

We have presented a forward-backward framework applicable to hidden CTSMC's. Avoidîng state space augmentation to enforce the Markov property, it gives rise to integral and integro-differential forward and backward equations and a scalable Viterbi algorithm. Adaptive step-size HSMM's can be derived, overcoming classical discrete-time core problems: out-of-grid observations and steady-state over-computation. We didn't consider CTSMC's that do not satisfy direction-time independence in this work. These are especially interesting for coarse-graining of large Markovian systems. Nonetheless, the algorithms can be generalized to this case by introduction of direction-dependent currents. Also, specific classes of waiting times could allow efficient computation of currents, e.g. power-law waiting times because of their relation to fractional calculus. Possible solutions would then fit naturally in our framework which handles observation likelihood of past events. Inference of model parameters, e.g. using an expectation maximization algorithm, can also be tackled using the forward algorithm from section 4.1.

**Acknowledgements and Funding Disclosure**

Nicolai Engelmann and Heinz Koeppl acknowledge support from the European Research Council (ERC) within the CONSYN project, grant agreement number 773196.

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
