# Tractable Latent State Inference for Hidden Continuous-Time semi-Markov Chains Supplement

Source Code at: https://anonymous.4open.science/r/BFghJAGII31

# Contents

# A  Derivations

We are considering a direction-time independent CTSMC $\{X(t)\}_{t \geq 0}$ on the discrete and finite state space $\mathcal{X}$. This does never change during the whole Appendix.

## A.1  Properties of Input-/Output Currents in the CTSMC (observation-free forward evolution)

### A.1.1  Infinitesimal In- and Out-Flow of Probability

The derivative of the state occupation marginal distribution $p(x, t)$ by time $t$ can be found from the joint probabilities of differing states with respect to two ends of a time-window of size $h$, i.e. $P(X(t) \neq x, X(t+h) = x)$ and $P(X(t) = x, X(t+h) \neq x)$ using the following reformulation

$$
\begin{aligned}
\frac{\mathrm{d}}{\mathrm{d}t} p(x, t) &= \lim_{h \to 0} \frac{P(X(t+h) = x) - P(X(t) = x)}{h} \\
&= \lim_{h \to 0} \frac{1}{h} \left( P(X(t) = x, X(t+h) = x) - P(X(t) = x, X(t+h) \neq x) \right. \\
&\quad \left. + P(X(t) \neq x, X(t+h) = x) - P(X(t) = x, X(t+h) = x) \right) \\
&= \lim_{h \to 0} \frac{1}{h} P(X(t) \neq x, X(t+h) = x) - \lim_{h \to 0} \frac{1}{h} P(X(t) = x, X(t+h) \neq x) \\
&= \phi(x, t) - \psi(x, t)
\end{aligned}
\tag{1}
$$

where we briefly refer to $\phi(x, t)$, the incoming infinitesimal probability flow in state $x$ at time instance $t$, by "input current of state $x$ at time $t$" and to $\psi(x, t)$, the outgoing infinitesimal flow, by "output current of state $x$ at time $t$" respectively. The quantity $\psi(x, t)$ occurres in the master equation of the inhomogenous CTMC in the same way as in the CTSMC and we can formulate $\frac{\mathrm{d}}{\mathrm{d}t} p(x, t)$ solely dependent on $\psi(x, t)$. This is, because

$$
\begin{aligned}
\phi(x, t) &= \lim_{h \to 0} \frac{1}{h} P(X(t) \neq x, X(t+h) = x) \\
&= \lim_{h \to 0} \frac{1}{h} \sum_{x' \neq x} P(X(t+h) = x, X(t) = x') \\
&= \sum_{x' \neq x} \lim_{h \to 0} \frac{1}{h} P(X(t+h) = x, X(t+h) \neq x', X(t) = x') \\
&= \sum_{x' \neq x} \lim_{h \to 0} \frac{1}{h} P(X(t) = x', X(t+h) \neq x') P(X(t+h) = x \mid X(t+h) \neq x', X(t) = x) \\
&= \sum_{x' \neq x} \lim_{h \to 0} \frac{1}{h} P(X(t) = x', X(t+h) \neq x') \lim_{h \to 0} P(X(t+h) = x \mid X(t+h) \neq x', X(t) = x) \\
&= \sum_{x' \neq x} \psi(x', t) m(x \mid x')
\end{aligned}
\tag{2}
$$

Thus, since $\sum_{x' \neq x} m(x' \mid x) = 1$, we have

$$\frac{\mathrm{d}}{\mathrm{d}t} p\left(x,\, t\right) = \sum_{x' \neq x} m\left(x \mid x'\right) \psi\left(x',\, t\right) - m\left(x' \mid x\right) \psi\left(x,\, t\right) \tag{3}$$

The infinitesimal quantities are generally referred to by "probability currents". In the literature, a balanced out version the currents referring to $\frac{\mathrm{d}}{\mathrm{d}t} p\left(x,\, t\right)$ directly as the "probability current in and out of state $x$" is common, but it is crucial for us to distinguish between the incoming and the outgoing part of the current, i.e. $\phi\left(x,\, t\right)$ and $\psi\left(x,\, t\right)$. We first derive the integral equations from the main text and then present the values of the currents if the chain is in a steady state.

### A.1.2 Integral Equations for Input and Output Currents

Like specified above, knowledge of the current $\psi\left(x,\, t\right)$ can completely describe the evolution of the marginals $p\left(x,\, t\right)$ given an initial condition $p\left(x,\, 0\right)$. The currents describe the occurrence of the time instances where the process has the Markov property. This can be seen when considering the sum

$$\sum_{x} \psi\left(x,\, t\right) h + o\left(h\right) = \sum_{x} P\left(X\left(t+h\right) \neq x,\, X\left(t\right) = x\right)$$
$$= P\left(X\left(t+h\right) \neq X\left(t\right)\right)$$

For the limit $h \to 0$, this converges to the probability of having a state change at $t$ and thus having the Markov property at this time instance. This circumstance allows us to use them later to do the forward-backward smoothing. We still don't know, how the currents $\phi\left(x,\, t\right)$ and $\psi\left(x,\, t\right)$ evolve. We can find this evolution by looking at trajectories between the time instances where the chain obeys the Markov property, i.e. between two state transitions. In the following, we use the now introduced notation for intervals. We sometimes wish to denote the event that $X\left(\tau\right) = \chi\left(\tau\right)$ follows some predefined trajectory $\forall \tau \in (t,\, t']$ which is given by some function $\chi\left(\tau\right)$. Then, there are two cases in this work. Either, we don't care about the function $\chi\left(\tau\right)$. Then, we simply write the event of $X\left(t\right)$ following a certain trajectory by use of the bold script $P\left(\mathbf{x}_{(t,\, t']}\right)$ with the interval $(t,\, t']$ as underscore. Or, we want $X\left(t\right)$ to follow a constant trajectory $\chi\left(\tau\right) = x = \mathrm{const.}$. Then, we write $P\left(\mathbf{x}_{(t,\, t']} = x\right)$. We now proceed with the derivations.

Let's say, the CTSMC enters state $x$ at time instance $t'$ (between $t'$ and $t' + h$) and leaves it again at time instance $t$ (between $t$ and $t + h$) irrespective of anything outside of the interval $(t',\, t]$. Then, if $N \equiv (t - t')/h$,

$$P\left(\mathbf{x}_{(t',\, t]} = x,\, X\left(t'\right) \neq x,\, X\left(t+h\right) \neq x\right) = \lim_{h \to 0} \frac{1}{h} P\left(X\left(t'\right) \neq x,\, X\left(t'+h\right) = x\right) \tag{4}$$
$$\times P\left(X\left(t' + (N+1)h\right) \neq x \mid X\left(t' + Nh\right) = x,\, X\left(t' + (N-1)h\right) = x,\, \dots\right)$$
$$\times \prod_{m=1}^{N-1} P\left(X\left(t' + (m+1)h\right) = x \mid X\left(t' + mh\right) = x,\, X\left(t' + (m-1)h\right) = x,\, \dots\right)$$
$$= \lambda\left(t - t',\, x\right) \Lambda\left(t - t' \mid x\right) \phi\left(x,\, t'\right)$$
$$= f\left(t - t' \mid x\right) \phi\left(x,\, t'\right),$$

where we remember the quantities $\lambda$ (exit rate), $\Lambda$ (survival function) and $f$ (probability density function) related to the waiting times in the main text. We can now obtain a relation between the history of $\phi\left(x,\, t\right)$ and the present $\psi\left(x,\, t\right)$ by marginalization of every possible path that started at $t - t'$ by entering state $x$ and lead to a change of state at time instance $t$. Thus, we have

$$\psi\left(x,\,t\right)=\int_{-\infty}^{t}\mathrm{d}t'\,P\left(X\left(t+h\right)\neq x,\,\mathbf{x}_{(t',\,t]}=x,\,X\left(t'\right)\neq x\right) \tag{5}$$

$$=\int_{-\infty}^{t}\mathrm{d}t'\,f\left(t-t'\mid x\right)\phi\left(x,\,t'\right)$$

$$=\int_{0}^{\infty}\mathrm{d}t'\,f\left(t'\mid x\right)\phi\left(x,\,t-t'\right)$$

Using the same rationale but not demanding a state change at the end of the time interval, i.e.

$$P\left(\mathbf{x}_{(t',\,t]}=x,\,X\left(t'\right)\neq x\right)=\lim_{h\to0}\frac{1}{h}P\left(X\left(t'\right)\neq x,\,X\left(t'+h\right)=x\right) \tag{6}$$

$$\times\prod_{m=1}^{N-1}P\left(X\left(t'+\left(m+1\right)h\right)=x\mid X\left(t'+mh\right)=x,\,X\left(t'+\left(m-1\right)h\right)=x,\,\dots\right)$$

$$=\Lambda\left(t-t'\mid x\right)\phi\left(x,\,t'\right)$$

we can marginalize over all paths which lead to the state $x$ at $t$ irrespective of what's afterwards. This recovers the marginals:

$$p\left(x,\,t\right)=\int_{-\infty}^{t}\mathrm{d}t'\,P\left(\mathbf{x}_{[t',\,t)}=x,\,X\left(t'\right)\neq x\right) \tag{7}$$

$$=\int_{-\infty}^{t}\mathrm{d}t'\,\Lambda\left(t-t'\mid x\right)\phi\left(x,\,t'\right)$$

$$=\int_{0}^{\infty}\mathrm{d}t'\,\Lambda\left(t'\mid x\right)\phi\left(x,\,t-t'\right)$$

We remember that we have shown above in (2) that $\phi\left(x,\,t\right)=\sum_{x'\neq x}m\left(x\mid x'\right)\psi\left(x',\,t\right)$. Under this knowledge, we first get the following identity for the output currents

$$\psi\left(x,\,t\right)=\int_{-\infty}^{t}\mathrm{d}t'\,f\left(t-t'\mid x\right)\sum_{x'\neq x}\theta\left(x\mid x'\right)\psi\left(x',\,t'\right) \tag{8}$$

$$=\sum_{x'\neq x}m\left(x\mid x'\right)\int_{-\infty}^{t}\mathrm{d}t'\,f\left(t-t'\mid x\right)\psi\left(x',\,t'\right),$$

and the following identity for the input currents $\phi\left(x,\,t\right)$

$$\phi\left(x,\,t\right)=\sum_{x'\neq x}m\left(x\mid x'\right)\int_{-\infty}^{t}\mathrm{d}t'\,f\left(t-t'\mid x'\right)\phi\left(x',\,t'\right).$$

Interestingly, the convolutional kernels of these systems of homogeneous Volterra integral equations of the second kind are given directly by the waiting time densities. We usually won't look back to an infinite past at any arbitrary time instance $t$ and therefor need one of several possible initial conditions. This then renders the integral equations "inhomogeneous". We first commence with the steady state currents.

### A.1.3 Input and Output Currents in the Steady State

Certain interesting properties of CTSMC's can be shown to hold when the process is in its steady state. There are many previous works dealing with deriving these properties, and to avoid distraction from the topic of interest, we only present the properties and refer to the relevant works where they are obtained. One very important property is that, given a state $x$ at a time $t$ for a CTSMC in equilibrium, the distribution of the time the process already lingers in this state, i.e. the time instance $t'$ the process has entered the current state $x$ at $t$, is simply given by [1]

$$P\left(\mathbf{x}_{(t',\,t]} = x,\, X\left(t'\right) \neq x \mid X\left(t\right) = x\right) = \bar{\lambda}\left(x\right)\Lambda\left(t - t' \mid x\right) \tag{9}$$

where $\bar{\lambda}\left(x\right) \equiv \mathsf{E}\left(f\left(\cdot \mid x\right)\right)^{-1}$ is the expected exit rate of state $x$ at equilibrium, i.e. the reciprocal of the first moment of the waiting time distribution in state $x$. Note, that (9) is a valid probability density over arrival times $t'$ at state $x$. With this knowledge, we can find the steady state values of the currents. At first, note

$$P\left(\mathbf{x}_{(t',\,t]} = x,\, X\left(t'\right) \neq x,\, X\left(t\right) = x\right) = \bar{\lambda}\left(x\right)\Lambda\left(t - t' \mid x\right)\bar{p}\left(x\right)$$

where $\bar{p}\left(x\right)$ is the state occupation marginal at equilibrium $\lim_{t\to\infty} p\left(x,\,t\right)$. Then, by simple comparison, we can see that

$$\Lambda\left(t - t' \mid x\right)\bar{\phi}\left(x,\,t'\right) = \Lambda\left(t - t' \mid x\right)\bar{\lambda}\left(x\right)\bar{p}\left(x\right)$$

which implies in this generality that $\bar{\phi}\left(x\right) \equiv \bar{\phi}\left(x,\,t'\right)$ is constant in time at equilibrium and is given by

$$\bar{\phi}\left(x\right) = \bar{\lambda}\left(x\right)\bar{p}\left(x\right)$$

Now consulting (5), we can find $\bar{\psi}\left(x,\,t\right)$, so that

$$\bar{\psi}\left(x,\,t\right) = \int_0^\infty \mathrm{d}t'\, f\left(t' \mid x\right)\bar{\phi}\left(x\right)$$
$$= \bar{\phi}\left(x\right) = \bar{\lambda}\left(x\right)\bar{p}\left(x\right).$$

Thus, also $\bar{\psi}\left(x\right) \equiv \bar{\psi}\left(x,\,t\right)$ is independent of time and equivalent to the input current $\bar{\phi}\left(x\right)$, which both are equal to $\bar{\lambda}\left(x\right)\bar{p}\left(x\right)$. Note, that alternatively from comparison with a CTMC and the definition of the output currents as $\psi\left(x,\,t\right) \equiv \lim_{h\to 0}\frac{1}{h}P\left(X\left(t\right) = x,\, X\left(t + h\right) \neq x\right) = \lim_{h\to 0}\frac{1}{h}P\left(X\left(t + h\right) \neq x \mid X\left(t\right) = x\right)P\left(X\left(t\right) = x\right)$, the equality $\bar{\psi}\left(x\right) = \bar{\lambda}\left(x\right)\bar{p}\left(x\right)$ can be deduced as well. By then seeing from (1) that the steady state must satisfy $\bar{\phi}\left(x\right) - \bar{\psi}\left(x\right) = 0$, we get the equality $\bar{\phi}\left(x\right) = \bar{\psi}\left(x\right) = \bar{\lambda}\left(x\right)\bar{p}\left(x\right)$.

### A.1.4 Optimal Markov Chain Approximation

The previously derived quantities $\bar{\lambda}\left(x\right)$ play an important role, when it comes to Markovian approximations of the CTSMC on the original state space $\mathcal{X}$. If we don't introduce additional hidden states and keep the original embedded Markov chain, the optimal continuous-time Markov chain approximation is given by the entries of the rate matrix $\boldsymbol{Q} \in \mathbb{R}^{|\mathcal{X}|\times|\mathcal{X}|}$, where

$$Q_{xx'} \equiv q\left(x,\, x'\right) \equiv \begin{cases} m\left(x \mid x'\right) \bar{\lambda}\left(x'\right) & x' \neq x \\ -\bar{\lambda}\left(x\right) & x' = x \end{cases}$$

correspond to the equilibrium exit rates of $\bar{\lambda}\left(x\right)$ of the original CTSMC. In this CTMC approximation the equilibrium marginals $\bar{p}\left(x\right)$ are preserved, so that the steady state of the original CTSMC and the one of the CTMC approximation are equivalent. We know from Markov chain theory [2] that given a marginal of current states $p\left(x, t\right)$, any future marginal $p\left(x, t'\right)$ can be obtained from the equation

$$\boldsymbol{p}\left(t'\right) = \exp\left(\boldsymbol{Q}\left(t' - t\right)\right) \boldsymbol{p}\left(t\right)$$

where $\boldsymbol{p}\left(t\right) \equiv \left(p\left(x_1, t\right), \ldots, p\left(x_N, t\right)\right)^{\mathrm{T}}$ is the vector of state marginals. $N$ does refer to the number of states, i.e. $N \equiv |\mathcal{X}|$ here. The matrix exponential $\exp\left(\boldsymbol{Q}\left(t' - t\right)\right)$ is the (forward-)propagator of the CTMC and has the semigroup property. From equation (3) we can deduce that

$$\frac{\mathrm{d}}{\mathrm{d}t} \boldsymbol{p}\left(t\right) = \boldsymbol{Q}\boldsymbol{p}\left(t\right) = \left(\boldsymbol{M} - \boldsymbol{I}\right) \boldsymbol{\psi}\left(t\right) \tag{10}$$

where $\boldsymbol{M}$ is the matrix representation of the transition operator M of the embedded Markov chain and $\boldsymbol{\psi}\left(t\right) \equiv \left(\psi\left(x_1, t\right), \ldots, \psi\left(x_N, t\right)\right)^{\mathrm{T}}$ is the vector of output currents. Recognizing $\mathrm{diag}\left(\bar{\boldsymbol{\lambda}}\right) \boldsymbol{p}\left(t\right) = \boldsymbol{\psi}\left(t\right)$ from section A.1.3 and left multiplying $\mathrm{diag}\left(\bar{\boldsymbol{\lambda}}\right)$ to (10) we obtain

$$\frac{\mathrm{d}}{\mathrm{d}t} \boldsymbol{\psi}\left(t\right) = \mathrm{diag}\left(\bar{\boldsymbol{\lambda}}\right) \left(\boldsymbol{M} - \boldsymbol{I}\right) \boldsymbol{\psi}\left(t\right)$$

Defining $\boldsymbol{R} \equiv \mathrm{diag}\left(\bar{\boldsymbol{\lambda}}\right) \left(\boldsymbol{M} - \boldsymbol{I}\right)$ we find the solution of the ODE to be

$$\boldsymbol{\psi}\left(t'\right) = \exp\left(\boldsymbol{R}\left(t' - t\right)\right) \boldsymbol{\psi}\left(t\right) \tag{11}$$

A similar relationship holds for $\boldsymbol{\phi}\left(t\right)$ and can analoguously be derived under consideration of (1). Let's briefly check if (11) satisfies the more general equation (8). We therefor write with $\boldsymbol{D}_f\left(\tau\right) \equiv \left(f\left(\tau \mid x_1\right), \ldots, f\left(\tau \mid x_N\right)\right)^{\mathrm{T}}$ the diagonal matrix of waiting time densities

$$\boldsymbol{\psi}\left(t\right) = \int_0^\infty \mathrm{d}\tau \, \boldsymbol{D}_f\left(\tau\right) \boldsymbol{M} \exp\left(\boldsymbol{R}\left(t - \tau\right)\right) \boldsymbol{\psi}\left(0\right)$$

$$= \int_0^\infty \mathrm{d}\tau \, \boldsymbol{D}_f\left(\tau\right) \boldsymbol{M} \exp\left(-\boldsymbol{R}\tau\right) \boldsymbol{\psi}\left(t\right).$$

Note, that not only $\exp\left(\boldsymbol{R}t\right) \exp\left(-\boldsymbol{R}\tau\right) = \exp\left(-\boldsymbol{R}\tau\right) \exp\left(\boldsymbol{R}t\right)$ commute, but that also $\boldsymbol{\psi}\left(t\right)$ is independent of the integration variable. Thus, to proof the identity, we need to show that

$$\boldsymbol{I} = \int_0^\infty \mathrm{d}\tau \, \boldsymbol{D}_f\left(\tau\right) \boldsymbol{M} \exp\left(-\boldsymbol{R}\tau\right).$$

Since in a CTMC we have exponential waiting times $\boldsymbol{D}_f(\tau) = \exp\left(-\text{diag}\left(\bar{\boldsymbol{\lambda}}\right)\tau\right)\text{diag}\left(\bar{\boldsymbol{\lambda}}\right)$ and by using the definition of $\boldsymbol{R}$ which allows us to write $\text{diag}\left(\bar{\boldsymbol{\lambda}}\right)\boldsymbol{M} \equiv \boldsymbol{R} + \text{diag}\left(\bar{\boldsymbol{\lambda}}\right)$ we have

$$\boldsymbol{I} = \int_0^\infty \mathrm{d}t \, \exp\left(-\text{diag}\left(\bar{\boldsymbol{\lambda}}\right)\tau\right)\boldsymbol{R}\exp\left(-\boldsymbol{R}\tau\right) + \exp\left(-\text{diag}\left(\bar{\boldsymbol{\lambda}}\right)\tau\right)\text{diag}\left(\bar{\boldsymbol{\lambda}}\right)\exp\left(-\boldsymbol{R}\tau\right),$$

from which, using the product rule of differentiation, we can find

$$\begin{aligned}
\boldsymbol{I} &= -\left[\exp\left(-\text{diag}\left(\bar{\boldsymbol{\lambda}}\right)\tau\right)\exp\left(-\boldsymbol{R}\tau\right)\right]_0^\infty \\
&= \boldsymbol{I},
\end{aligned}$$

which verifies (11).

### A.1.5 Boundary Conditions for Input and Output Currents

In this section, we introduce boundary conditions. Any of these conditions introduces an inhomogeneity $g$ into equation (5) and thus the two derived identities. We start with cutting the integration into two of ranges $(-\infty, t')$ and $[t', t)$ which gives us

$$\psi(x, t) = \int_{-\infty}^{t'} \mathrm{d}\tau \, f(t - \tau \mid x)\,\phi(x, \tau) + \int_{t'}^t \mathrm{d}\tau \, f(t - \tau \mid x)\,\phi(x, \tau)$$

We choose $t'$ so that we have no knowledge of the part of the trajectory in $(-\infty, t')$. Thus, we assume it to be a general function of $t$, so that we need to look at the inhomogeneous equations

$$\psi(x, t) = g(x, t', t) + \int_{t'}^t \mathrm{d}\tau \, f(t - \tau \mid x)\,\phi(x, \tau), \tag{12}$$

where $g$ assumes the following forms depending on the boundary conditions. In matrix-vector notation, the equation reads

$$\boldsymbol{\psi}(t) = \boldsymbol{g}(t', t) + \int_{t'}^t \mathrm{d}\tau \, \boldsymbol{D}_f(t - \tau)\,\boldsymbol{\phi}(\tau)$$

Without loss of generality, we now assume $t' = 0$. We consider the following initial conditions also considered in [3].

- There is a transition immediately at the beginning of the considered time interval $[0, t)$. This means, that $\forall \tau < 0$: $\phi(x, \tau) \equiv \delta(\tau)\,p(x, 0)$, which leads to

$$\begin{aligned}
g(x, t) &\equiv \int_t^\infty \mathrm{d}\tau \, f(\tau \mid x)\,\delta(t - \tau)\,p(x, 0) \\
&= f(t \mid x)\,p(x, 0)
\end{aligned}$$

- The process is assumed to be in steady-state before the beginning of the considered time interval $[0, t)$. This means, that $\forall \tau < 0 : \phi(x, \tau) \equiv \bar{\phi}(x) = \bar{\psi}(x) = \bar{\lambda}(x)\bar{p}(x)$ (for the equality, rf. to A.1.3), which leads to

$$
\begin{aligned}
g(x, t) &\equiv \int_t^\infty \mathrm{d}\tau\, f(\tau \mid x)\bar{\phi}(x) \\
&= (1 - F(t \mid x))\bar{\phi}(x) \\
&= \Lambda(t \mid x)\bar{\phi}(x) \\
&= \Lambda(t \mid x)\bar{\lambda}(x)\bar{p}(x)
\end{aligned}
$$

considering the inhomogeneous equation (12) and the two derived identities under inclusion of the inhomogeneities, we can uniquely solve for the currents $\phi(x, t)$ and $\psi(x, t)$ and obtain the marginals $p(x, t)$ by solving a system of inhomogeneous Volterra integral equations of the second kind. For this, there are several solution strategies available []. Next, we take a look what happens if we introduce point-wise evidence.

## A.2 Forward and Backward Terms in the CTSMC

### A.2.1 Observation Model and Likelihood Normalization

We now assume, we are given a sequence of $K$ measurements and measurement times $\mathcal{E} \equiv \{(y_k, t_k) \mid k \in \{0, 1, \ldots, K\}\}$, where the measurement indices are in ascending order, so $t_0 < t_1 < \cdots < t_K$. For convenience, we introduce the parametrized subset $\mathcal{E}(t, t') \equiv \{(y_k, t_k) \in \mathcal{E} \mid t_k \in (t, t']\}$, which returns all measurements within the time interval $(t, t']$. Please also review section 1.2 in the main text.

where $\check{v} : \mathbb{R} \times \mathbb{R} \to \mathbb{R}_{\geq 0}$ is a piecewise constant function both in its first and second argument, which returns the likelihood of all observations made in the time-interval $(t', t]$ given state $x$, i.e.

$$
\check{v}(x, t', t) \equiv \prod_{(y', \tau) \in \mathcal{E}[t, t')} P(Y(\tau) = y' \mid X(\tau) = x)
$$

We define the normalized joint observation likelihood function by

$$
v(x, t', t) \equiv \prod_{(y, \tau) \in \mathcal{E}[t', t)} \frac{P(Y(\tau) = y \mid X(\tau) = x)}{P(Y(\tau) = y \mid \mathbf{y}_{(0, \tau]})}
$$

### A.2.2 Definition of Forward Terms

To begin this section, we present the definitions of forward and backward currents. We first define the unnormalized forward or filtered marginals $\check{\alpha}(x, t)$ by

$$
\check{\alpha}(x, t) \equiv P\left(X(t) = x, \bigcap_{(y, \tau) \in \mathcal{E}(0, t)} Y(\tau) = y\right)
$$

In analogy, we then define the unnormalized forward currents $\check{\phi}_\alpha(x, t)$, $\check{\psi}_\alpha(x, t)$ as

$$\check{\psi}_\alpha\left(x,\,t\right) \equiv P\left(X\left(t\right) = x,\, X\left(t+h\right) \neq x,\, \bigcap_{(y,\,\tau)\in\mathcal{E}(0,\,t)} Y\left(\tau\right) = y\right)$$

$$\check{\phi}_\alpha\left(x,\,t\right) \equiv P\left(X\left(t\right) \neq x,\, X\left(t+h\right) = x,\, \bigcap_{(y,\,\tau)\in\mathcal{E}(0,\,t)} Y\left(\tau\right) = y\right) = \boldsymbol{M}\check{\psi}_\alpha\left(x,\,t\right)$$

Note, that the last equality is directly given by (2) by simply including the joint event of the past evidence in the single terms. At this point, we introduce the notation $\mathbf{y}_{[0,\,t)} \equiv \left\{\bigcap_{(y,\,\tau)\in\mathcal{E}(0,\,t)} Y\left(\tau\right) = y\right\}$ for the joint event of all observations in the time window $(0,\,t]$. The normalized terms are then given by division by duration-normalized probability of the evidence, i.e.

$$\alpha\left(x,\,t\right) \equiv \frac{\check{\alpha}\left(x,\,t\right)}{P\left(\mathbf{y}_{(0,\,t]}\right)} = P\left(X\left(t\right) = x \,\middle|\, \bigcap_{(y,\,\tau)\in\mathcal{E}(0,\,t)} Y\left(\tau\right) = y\right)$$

$$\psi_\alpha\left(x,\,t\right) \equiv \frac{\check{\psi}_\alpha\left(x,\,t\right)}{P\left(\mathbf{y}_{(0,\,t]}\right)} = P\left(X\left(t\right) = x,\, X\left(t+h\right) \neq x \,\middle|\, \bigcap_{(y,\,\tau)\in\mathcal{E}(0,\,t)} Y\left(\tau\right) = y\right)$$

$$\phi_\alpha\left(x,\,t\right) \equiv \frac{\check{\phi}_\alpha\left(x,\,t\right)}{P\left(\mathbf{y}_{(0,\,t]}\right)} = P\left(X\left(t\right) \neq x,\, X\left(t+h\right) = x \,\middle|\, \bigcap_{(y,\,\tau)\in\mathcal{E}(0,\,t)} Y\left(\tau\right) = y\right) = \boldsymbol{M}\psi_\alpha\left(x,\,t\right)$$

For the last equality, again note neither intersecting with nor conditioning on the past observations invalidates (2).

### A.2.3   Forward Integral Equations under Observations

Following the derivations in A.1.2, we have for the discrete time limit for a single path from entering state $x$ at time instance $t'$ to exiting at time instance $t$ under observations

$$P\left(\mathbf{x}_{(t',\,t]} = x,\, X\left(t\right) \neq x,\, \bigcap_{(y',\,\tau)\in\mathcal{E}(0,\,t)} Y\left(\tau\right) = y'\right) = \prod_{(y',\,\tau)\in\mathcal{E}(t',\,t)} P\left(Y\left(\tau\right) = y' \mid X\left(\tau\right) = x\right)$$

$$\times P\left(\mathbf{x}_{(t',\,t]} = x,\, X\left(t\right) \neq x,\, \bigcap_{(y',\,\tau)\in\mathcal{E}(0,\,t')} Y\left(\tau\right) = y'\right)$$

$$= \check{\upsilon}\left(x,\,t',\,t\right) f\left(t - t' \mid x\right) \phi_\alpha\left(x,\,t'\right)$$

Because of the Markov property of the observation, conditioning on the state $X\left(\tau\right)$ renders the observations independent. Thus, the joint observation likelihood for the time interval is simply the product of the single observation likelihoods contained in the interval. Similar to the version without observations,

$$\psi_\alpha(x, t) = \int_0^\infty dt' \lim_{h \to 0} \frac{1}{h} P\left( X(t') \neq x,\, X(t+h) \neq x,\, \mathbf{x}_{(t', t]} = x \,\Big|\, \mathbf{y}_{(0, t]} \right) \tag{13}$$

$$= \int_0^\infty dt' \lim_{h \to 0} \frac{1}{h} \frac{P\left( X(t') \neq x,\, X(t+h) \neq x,\, \mathbf{x}_{(t', t]} = x,\, \mathbf{y}_{(t', t]} \,\Big|\, \mathbf{y}_{(0, t']} \right)}{P\left( \mathbf{y}_{(t', t]} \mid \mathbf{y}_{(0, t']} \right)}$$

$$= \int_0^\infty dt' \lim_{h \to 0} \frac{1}{h} \frac{\prod_{(y', \tau) \in \mathcal{E}(t', t)} P(Y(\tau) = y' \mid X(\tau) = x)}{P\left( \mathbf{y}_{(t', t]} \mid \mathbf{y}_{(0, t']} \right)} P\left( X(t') \neq x,\, X(t+h) \neq x,\, \mathbf{x}_{(t', t]} = x \,\Big|\, \mathbf{y}_{(0, t']} \right)$$

$$= \int_0^\infty dt' \prod_{(y, \tau) \in \mathcal{E}(t', t)} \frac{P(Y(\tau) = y \mid X(\tau) = x)}{P(Y(\tau) = y \mid \mathbf{y}_{(0, \tau]})}$$

$$\times \lim_{h \to 0} \frac{1}{h} P\left( X(t) = x,\, X(t+h) \neq x,\, \mathbf{x}_{(t', t]} = x \,\Big|\, X(t') \neq x,\, X(t'+h) = x \right)$$

$$\times P\left( X(t') \neq x,\, X(t'+h) = x \,\Bigg|\, \bigcap_{(y, \tau) \in \mathcal{E}(0, t')} Y(\tau) = y \right)$$

$$= \int_0^\infty dt'\, v(x, t', t)\, f(t - t' \mid x)\, \phi_\alpha(x, t')$$

where we refer to (4) for the middle term in the second last line. And by again considering (2), we also obtain the two equations

$$\psi_\alpha(x, t) = \sum_{x' \neq x} m(x \mid x') \int_0^\infty dt'\, f(t - t' \mid x')\, \psi_\alpha(x', t'), \tag{14}$$

and for the input forward currents $\phi_\alpha(x, t)$

$$\phi_\alpha(x, t) = \sum_{x' \neq x} m(x \mid x') \int_0^\infty dt'\, f(t - t' \mid x')\, \phi_\alpha(x', t').$$

The previous equations can also be written in a matrix-vector form and an operator form. Depending on the context, each formulation is more or less appropriate. For the matrix-vector notation, we introduce the diagonal observation likelihood matrix $\boldsymbol{D}_v(t', t) \equiv (v(x_1, t', t), \ldots, v(x_N, t', t))^{\mathrm{T}}$. For the operator notation, we introduce the integral operator $\mathsf{F}_\alpha$, which acts on a suitable test-function $g$ by $\mathsf{F}_\alpha g \equiv \int_0^t dt'\, v(x, t', t)\, f(t - t' \mid x)\, g(x, t')$. We then have

$$\psi_\alpha = \mathsf{F}_\alpha \phi_\alpha + v_0 g_\psi \qquad \Leftrightarrow \qquad \boldsymbol{\psi}_\alpha(t) = \int_0^\infty dt'\, \boldsymbol{D}_v(t', t)\, \boldsymbol{D}_f(t - t')\, \boldsymbol{\phi}_\alpha(t')$$

$$\psi_\alpha = \mathsf{F}_\alpha \mathsf{M} \psi_\alpha + v_0 \mathsf{M} g_\psi \qquad \Leftrightarrow \qquad \boldsymbol{\psi}_\alpha(t) = \int_0^\infty dt'\, \boldsymbol{D}_v(t', t)\, \boldsymbol{D}_f(t - t')\, \boldsymbol{M} \boldsymbol{\psi}_\alpha(t')$$

$$\phi_\alpha = \mathsf{M} \mathsf{F}_\alpha \phi_\alpha + \mathsf{M} v_0 g_\psi \qquad \Leftrightarrow \qquad \boldsymbol{\phi}_\alpha(t) = \int_0^\infty dt'\, \boldsymbol{M} \boldsymbol{D}_v(t', t)\, \boldsymbol{D}_f(t - t')\, \boldsymbol{\phi}_\alpha(t')$$

### A.2.4 Definition of Backward Terms

We have introduced the forward input and output currents but it is also possible to formulate a current model for the likelihood of a future path. We call these currents the backward input and output currents and they represent the instantaneous flow of future observation likelihood in and out of a state. We define the unnormalized backward "marginals" $\check{\beta}$ by

$$\check{\beta}(x,\,t) \equiv P\left(\bigcap_{(y,\,\tau)\in\mathcal{E}(t,\,T)} Y(\tau) = y,\, X(t) = x\right) = P\left(\mathbf{y}_{(t,\,T]},\, X(t) = x\right)$$

In section (X) we will give a justification for the naming of $\check{\beta}$ as "marginals", since they are obtained in the form of a marginal likelihood. In the following, we will thus dispense with the quotation marks. We then also define the unnormalized backward currents $\phi_\beta(x,\,t)$ and $\psi_\beta(x,\,t)$ by

$$\phi_\beta(x,\,t) = \lim_{h\to 0} P\left(\mathbf{y}_{(t,\,T]} \mid X(t) \neq x,\, X(t+h) = x\right)$$
$$\psi_\beta(x,\,t) = \lim_{h\to 0} P\left(\mathbf{y}_{(t,\,T]} \mid X(t) = x,\, X(t+h) \neq x\right)$$

Before we talk about normalization, we obtain a relationshipt between $\phi_\beta(x,\,t)$ and $\psi_\beta(x,\,t)$ similar to (2). First, we need to see that the following likelihoods are identical:

$$\lim_{h\to 0} P\left(\mathbf{y}_{(t,\,T]},\, \mid X(t+h) = x',\, X(t+h) \neq x,\, X(t) = x\right) = \lim_{h\to 0} P\left(\mathbf{y}_{(t,\,T]},\, \mid X(t+h) = x',\, X(t) = x\right)$$
$$= \lim_{h\to 0} P\left(\mathbf{y}_{(t,\,T]},\, \mid X(t+h) = x',\, X(t) \neq x'\right)$$

The first equality can be explained by the enclosure property of the left condition within the right. The second equality, however, is something specific to the renewal property of CTSMC's. Since the process has the Markov property at the time of transition, the likelihood of future observations does not change with the state we transitioned from. Using both properties, we can find

$$\begin{aligned}
\psi_\beta(x,\,t) &= P\left(\mathbf{y}_{(t,\,T]},\, \mid X(t+h) \neq x,\, X(t) = x\right) \\
&= \sum_{x'\neq x} P\left(\mathbf{y}_{(t,\,T]},\, X(t+h) = x' \mid X(t+h) \neq x,\, X(t) = x\right) \\
&= \sum_{x'\neq x} P\left(\mathbf{y}_{(t,\,T]},\, \mid X(t+h) = x',\, X(t+h) \neq x,\, X(t) = x\right) P\left(X(t+h) = x' \mid X(t+h) \neq x,\, X(t) = x\right) \\
&= \sum_{x'\neq x} P\left(\mathbf{y}_{(t,\,T]},\, \mid X(t+h) = x',\, X(t) = x\right) P\left(X(t+h) = x' \mid X(t+h) \neq x,\, X(t) = x\right) \\
&= \sum_{x'\neq x} P\left(\mathbf{y}_{(t,\,T]},\, \mid X(t+h) = x',\, X(t) \neq x'\right) P\left(X(t+h) = x' \mid X(t+h) \neq x,\, X(t) = x\right) \\
&= \sum_{x'\neq x} m\left(x' \mid x\right) \phi_\beta\left(x',\,t\right) \quad\quad\quad\quad\quad\quad (15)
\end{aligned}$$

If a normalization is applied to the backward terms is problem-specific and sometimes even a matter of choice. However, in practice, we have experienced that the backward currents vanish quickly. Thus, a normalization is useful or even necessary when inference is done over longer time windows. As a normalization factor, we chose the likelihood of the past data with respect to the future data, i.e. $P\left(\mathbf{y}_{(t',\,t]}\mid\mathbf{y}_{(0,\,t']}\right)$. For the normalizers to obtain, we thus first need to calculate the forward quantities. However, since a backward pass is usually performed in conjunction with a forward pass for Bayesian smoothing, we can handover the obtained values if the forward pass is performed first. Then, we write for the normalized quantities

$$\beta\left(x,\,t\right) \equiv \frac{P\left(\mathbf{y}_{(t,\,T]},\,X\left(t\right)=x\right)}{P\left(\mathbf{y}_{(t,\,T]}\mid\mathbf{y}_{(0,\,t]}\right)}$$

$$\psi_\beta\left(x,\,t\right) \equiv \frac{P\left(\mathbf{y}_{(t,\,T]},\,X\left(t+h\right)\neq x,\,X\left(t\right)=x\right)}{P\left(\mathbf{y}_{(t,\,T]}\mid\mathbf{y}_{(0,\,t]}\right)} \qquad \phi_\beta\left(x,\,t\right) \equiv \frac{P\left(\mathbf{y}_{(t,\,T]},\,X\left(t+h\right)=x,\,X\left(t\right)\neq x\right)}{P\left(\mathbf{y}_{(t,\,T]}\mid\mathbf{y}_{(0,\,t]}\right)}$$

Their natural time arrow is backwards and thus the integral equations obtained in the next section will be dependent on values of these currents at future time instances.

### A.2.5 Backward Integral Equations under Observations

To derive the fundamental integral equation relating the backward currents the same rationale behind obtaining the forward equations can be applied. First, consider the likelihood

$$\lim_{h\to 0}\frac{1}{h}P\left(\mathbf{y}_{(t,\,T]},\,X\left(t+\tau+h\right)\neq x,\,X\left(t+\tau\right)=x,\,\mathbf{x}_{(t,\,t+\tau]}=x\mid X\left(t+h\right)=x,\,X\left(t\right)\neq x\right)$$
$$=\lim_{h\to 0}P\left(\mathbf{y}_{(t+\tau,\,T]},\mid X\left(t+\tau+h\right)\neq x,\,X\left(t+\tau\right)=x\right)P\left(\mathbf{y}_{(t,\,t+\tau]}\mid\mathbf{x}_{(t,\,t+\tau]}=x\right)$$
$$\times\frac{1}{h}P\left(X\left(t+\tau+h\right)\neq x,\,X\left(t+\tau\right)=x,\,\mathbf{x}_{(t,\,t+\tau]}=x\mid X\left(t+h\right)=x,\,X\left(t\right)\neq x\right)$$
$$=\check{v}\left(x,\,t,\,t+\tau\right)\check{\psi}_\beta\left(x,\,t+\tau\right)f\left(\tau\right)$$

which gives the likelihood of a state change at $t$ to state $x$ w.r.t. the event that we stay for duration $\tau$ in state $x$, observe $\mathbf{y}_{(t,\,t+\tau]}$ within that time window, and then leave $x$ to any other state $x'\neq x$ and observe $\mathbf{y}_{(t+\tau,\,T]}$ in the future. Note, that we used the unnormalized $\check{v}$ and $\check{\psi}_\beta$. Introducing the normalizer $P\left(\mathbf{y}_{(t,\,T]}\mid\mathbf{y}_{(0,\,t]}\right)=P\left(\mathbf{y}_{(t+\tau,\,T]}\mid\mathbf{y}_{(0,\,t+\tau]}\right)P\left(\mathbf{y}_{(t,\,t+\tau]}\mid\mathbf{y}_{(0,\,t]}\right)$ to the equation then yields

$$\lim_{h\to 0}\frac{P\left(\mathbf{y}_{(t+\tau,\,T]},\mid X\left(t+\tau+h\right)\neq x,\,X\left(t+\tau\right)=x\right)}{P\left(\mathbf{y}_{(t+\tau,\,T]}\mid\mathbf{y}_{(0,\,t+\tau]}\right)}\frac{P\left(\mathbf{y}_{(t,\,t+\tau]}\mid\mathbf{x}_{(t,\,t+\tau]}=x\right)}{P\left(\mathbf{y}_{(t,\,t+\tau]}\mid\mathbf{y}_{(0,\,t]}\right)}$$
$$\times\frac{1}{h}P\left(X\left(t+\tau+h\right)\neq x,\,X\left(t+\tau\right)=x,\,\mathbf{x}_{(t,\,t+\tau]}=x\mid X\left(t+h\right)=x,\,X\left(t\right)\neq x\right)$$
$$=v\left(x,\,t,\,t+\tau\right)\psi_\beta\left(x,\,t+\tau\right)f\left(\tau\right)$$

Again, like in (13), we used the idea that we have precise knowledge of $P\left(\mathbf{y}_{(t,\,t+\tau]}\mid\mathbf{x}_{(t,\,t+\tau]}=x\right)=\prod_{(y,\,s)\in\mathcal{E}(t,\,t+\tau)}P\left(Y\left(s\right)=y\mid$ Thus, using again the renewal property of the CTSMC, we can perform a marginalization over all possible future paths by integrating along $\tau$, i.e.

$$\phi_\beta(x,\, t) = \lim_{h\to 0} P\left(\mathbf{y}_{(t,\, T]},\, |\, X(t+h) = x,\, X(t) \neq x\right)$$

$$= \int_0^\infty d\tau\, P\left(\mathbf{y}_{(t,\, T]},\, X(t+\tau) \neq x,\, \mathbf{x}_{(t,\, t+\tau]} = x\, |\, X(t+h) = x,\, X(t) \neq x\right)$$

$$= \int_0^\infty d\tau\, \upsilon(x,\, t,\, t+\tau)\, \psi_\beta(x,\, t+\tau)\, f(\tau\, |\, x)$$

$$= \int_t^T d\tau\, \upsilon(x,\, t,\, \tau)\, \psi_\beta(x,\, \tau)\, f(\tau - t\, |\, x) + \gamma_\psi(x,\, t) \tag{16}$$

with the inhomogeneity $\gamma_\psi : \mathcal{X} \times \mathbb{R}_{\geq 0} \to \mathbb{R}_{\geq 0}$ implementing the right boundary conditions.

We already mentioned the future likelihood marginals $\beta(x,\, t) \equiv P\left(\mathbf{y}_{(t,\, T]}\, |\, X(t) = x\right)$. We will now derive an expression for $\beta(x,\, t)$ in terms of future currents $\psi_\beta(x,\, t)$ similar to (5) and then find and expression ot the time-derivative $\frac{d}{dt}\beta(x,\, t)$. We will need both later to derive the backward equation. To derive the first equation resolving $q(x,\, t)$, we basically follow the derivation of (16) and examine an intermediate point. Precisely, we look at

$$\lim_{h\to 0} \frac{1}{h} P\left(\mathbf{y}_{(t,\, T]}\, |\, X(t) = x\right)$$

$$= \int_0^{T-t} d\tau \int_0^\infty d\tau'\, \lim_{h\to 0} P\left(\mathbf{y}_{(t+\tau,\, T]},\, |\, X(t+\tau+h) \neq x,\, X(t+\tau) = x\right)$$

$$\times P\left(\mathbf{y}_{(t,\, t+\tau]},\, X(t+\tau+h) \neq x,\, X(t+\tau) = x,\, \mathbf{x}_{(t,\, t+\tau]} = x\, |\, \mathbf{x}_{(t-\tau',\, t]} = x,\, X(t-\tau'+h) = x,\, X(t-\tau') \neq x\right)$$

$$\times P\left(\mathbf{x}_{(t-\tau',\, t]} = x\, |\, X(t-\tau'+h) = x,\, X(t-\tau') \neq x\right) P\left(X(t-\tau'+h) = x,\, X(t-\tau') \neq x\right) \tag{17}$$

The rationale is the following. To obtain $\beta(x,\, t)$, we need to marginalize over all paths that lead to a residence in state $x$ at time instance $t$. So, the only thing we know is that entry into state $x$ must happen at a time $t - \tau' \leq t$ and exit must happen at a time $t + \tau > t$. The only unknown in this equation is $P\left(X(t-\tau'+h) = x,\, X(t-\tau') \neq x\right)$. Since we are considering a fully unconditional probability of state change, we assume the process to be in steady state (rf. section (A.1.3)). Thus, we assume a uniform - i.e. uninformed - density across state change times and state occupation. This means that we must insert the steady-state value for $P\left(X(t-\tau'+h) = x,\, X(t-\tau') \neq x\right)$ but under the assumption of equally probable state occupation $P\left(X(t-\tau'+h) = x\right) = |\mathcal{X}|^{-1}$. Then,

$$\beta(x,\, t) = \int_0^\infty d\tau \int_0^\infty d\tau'\, \bar{\lambda}(x)\, f(\tau' + \tau\, |\, x)\, \upsilon(x,\, t,\, t+\tau)\, \psi_\beta(x,\, t+\tau)$$

$$= \int_0^{T-t} d\tau\, \bar{\lambda}(x)\, \Lambda(\tau\, |\, x)\, \upsilon(x,\, t,\, t+\tau)\, \psi_\beta(x,\, t+\tau) + \upsilon(x,\, t,\, T)\, \gamma_q(x,\, t)$$

$$= \int_t^T d\tau\, \bar{\lambda}(x)\, \Lambda(\tau - t\, |\, x)\, \upsilon(x,\, t,\, \tau)\, \psi_\beta(x,\, \tau) + \upsilon(x,\, t,\, T)\, \gamma_q(x,\, t) \tag{18}$$

which uncoincidently bears a strong resemblence to (7). The term $\gamma_q : \mathcal{X} \times \mathbb{R}_{\geq 0} \to \mathbb{R}_{\geq 0}$ again denotes an inhomogeneity encoding the right boundary conditions. By deriving this equation by $t$, we can also find a relationship between the derivative $\frac{d}{dt}\beta(x,\, t)$ and the two backward currents $\psi_\beta(x,\, t)$ and $\phi_\beta(x,\, t)$ similar to (1). Taking the derivative yields

$$\frac{\mathrm{d}}{\mathrm{d}t}\beta\left(x,\,t\right)=-\bar{\lambda}\left(x\right)\Lambda\left(0\right)\psi_{\beta}\left(x,\,\tau\right)+\bar{\lambda}\left(x\right)\int_{t}^{T}\mathrm{d}\tau\,\upsilon\left(x,\,t,\,\tau\right)f\left(\tau-t\mid x\right)\psi_{\beta}\left(x,\,\tau\right)$$

$$=\bar{\lambda}\left(x\right)\left[\phi_{\beta}\left(x,\,t\right)-\psi_{\beta}\left(x,\,t\right)\right]$$

$$=\bar{\lambda}\left(x\right)\left[\phi_{\beta}\left(x,\,t\right)-\sum_{x'\neq x}m\left(x'\mid x\right)\phi_{\beta}\left(x',\,t\right)\right] \tag{19}$$

where in the last line we used result (15).

## A.3   Kolmogorov Equations (self-contained evolution of forward/backward terms)

In this section, we present the derivations for the Kolmogorov equations and extend them to account for evidence.

### A.3.1   Classical Forward/Master Equation

We now derive the classical observation-free forward equation consistent with the formulations in this work. In the following $\mathcal{L}$ will denote the Laplace transform operator with its inverse $\mathcal{L}^{-1}$ and the Laplace transform of a known function will be denoted by a tilde, so e.g. $\tilde{f}\left(x,\,s\right)=\mathcal{L}\left(f\left(x,\,\cdot\right)\right)$. We start out by remembering equation 5. There, the past input current is related to the output current at time $t$ for any state $x$. We also remember that in equation 12, we let boundary conditions enter the equation in the form of an inhomogeneity $g\left(x,\,t\right)$. Then, applying the Laplace transform (as written applied to the time dimension) on 3, we have

$$s\tilde{p}\left(x,\,s\right)-p\left(x,\,0^{-}\right)=\sum_{x'\neq x}m\left(x\mid x'\right)\tilde{\psi}\left(x',\,s\right)-m\left(x'\mid x\right)\tilde{\psi}\left(x,\,s\right)$$

the inserting the Laplace transform of 12, we get

$$s\tilde{p}\left(x,\,s\right)=\sum_{x'\neq x}m\left(x\mid x'\right)\left(\tilde{f}\left(x',\,s\right)\tilde{\phi}\left(x',\,s\right)+\tilde{g}_{\psi}\left(x',\,s\right)\right)$$
$$-m\left(x'\mid x\right)\left(\tilde{f}\left(x,\,s\right)\tilde{\phi}\left(x,\,s\right)+\tilde{g}_{\psi}\left(x,\,s\right)\right)$$

and then, from the Laplace transform of (7), $\tilde{p}\left(x,\,s\right)=\tilde{\Lambda}\left(x,\,s\right)\tilde{\phi}\left(x,\,s\right)+\tilde{g}_{p}\left(x,\,s\right)$, we get

$$s\tilde{p}\left(x,\,s\right)-p\left(x,\,0^{-}\right)=\sum_{x'\neq x}m\left(x\mid x'\right)\left(\frac{\tilde{f}\left(x',\,s\right)}{\tilde{\Lambda}\left(x',\,s\right)}\left(\tilde{p}\left(x',\,s\right)-\tilde{g}_{p}\left(x',\,s\right)\right)+\tilde{g}_{\psi}\left(x',\,s\right)\right)$$
$$-m\left(x'\mid x\right)\left(\frac{\tilde{f}\left(x,\,s\right)}{\tilde{\Lambda}\left(x,\,s\right)}\left(\tilde{p}\left(x,\,s\right)-\tilde{g}_{p}\left(x,\,s\right)\right)+\tilde{g}_{\psi}\left(x,\,s\right)\right)$$

Inverting the Laplace transform, we arrive at a general form of the forward equation, which is

$$\frac{\mathrm{d}}{\mathrm{d}t}p\left(x,\,t\right)=\sum_{x'\neq x}m\left(x\mid x'\right)\int_0^t\mathrm{d}\tau\,\kappa\left(x',\,t-\tau\right)p\left(x',\,\tau\right)-m\left(x'\mid x\right)\int_0^t\mathrm{d}\tau\,\kappa\left(x,\,t-\tau\right)p\left(x,\,\tau\right)+g\left(x,\,t\right) \qquad (20)$$

with the memory kernel $\kappa\left(x,\,t\right)\equiv\mathcal{L}^{-1}\left(\frac{\tilde{f}(x,\,s)}{\tilde{\Lambda}(x,\,s)}\right)$ and the inhomogeneity

$$g\left(x,\,t\right)=\sum_{x'\neq x}m\left(x\mid x'\right)\left(g_\psi\left(x',\,t\right)-\int_0^t\mathrm{d}\tau\,\kappa\left(x',\,t-\tau\right)g_p\left(x',\,\tau\right)\right)-m\left(x'\mid x\right)\left(g_\psi\left(x,\,t\right)-\int_0^t\mathrm{d}\tau\,\kappa\left(x,\,t-\tau\right)g_p\left(x,\,\tau\right)\right).$$

Remembering that $\sum_{x'\neq x}m\left(x'\mid x\right)=1$, we can get additional structural detail from matrix-vector notation, which gives

$$\frac{\mathrm{d}}{\mathrm{d}t}\boldsymbol{p}\left(t\right)=\left(\boldsymbol{M}-\boldsymbol{I}\right)\int_0^t\mathrm{d}\tau\,\boldsymbol{K}\left(t-\tau\right)\boldsymbol{p}\left(\tau\right)+\boldsymbol{g}\left(t\right)$$

and for the inhomogeneity

$$\boldsymbol{g}\left(t\right)=\left(\boldsymbol{M}-\boldsymbol{I}\right)\left(\boldsymbol{g}_\psi\left(t\right)-\int_0^t\mathrm{d}\tau\,\boldsymbol{K}\left(t-\tau\right)\boldsymbol{g}_p\left(\tau\right)\right)$$

Usually, the equation is derived for a specific boundary condition under which $\boldsymbol{g}\left(t\right)$ can be omitted []. This either means that $\boldsymbol{g}\left(t\right)=0$ or $\boldsymbol{g}\left(t\right)\in\mathrm{null}\left(\boldsymbol{M}-\boldsymbol{I}\right)$, while the latter in the general case means $g\left(x,\,t\right)=\bar{\lambda}\left(x\right)\bar{p}\left(x\right)$. If we assume a transition to occur at $t=0$, we can achieve the case that $\boldsymbol{g}\left(t\right)=0$. This would reflect in the inhomogeneities as (rf. section A.1.5)

$$\tilde{g}_p\left(x,\,s\right)=\tilde{\Lambda}\left(x,\,s\right)\bar{p}\left(x\right)$$
$$\tilde{g}_\psi\left(x,\,s\right)=\tilde{f}\left(x,\,s\right)\bar{p}\left(x\right).$$

Then

$$\frac{\tilde{f}\left(x,\,s\right)}{\tilde{\Lambda}\left(x,\,s\right)}\tilde{g}_p\left(x,\,s\right)=\tilde{f}\left(x,\,s\right)\bar{p}\left(x\right)$$

and so

$$\tilde{g}_\psi\left(x,\,s\right)-\frac{\tilde{f}\left(x,\,s\right)}{\tilde{\Lambda}\left(x,\,s\right)}\tilde{g}_p\left(x,\,s\right)=0,$$

which removes the inhomogeneous term entirely, i.e. $g\left(x,\,t\right)=0$. In this work, we need the general version 20 with unspecified inhomogeneities to do the forward and backward sweeps.

## A.3.2  Forward Equation under Observations

We have shown in 12 that we can decompose an integral equation on a domain $(0,\,T]$ into two separate equations on $(0,\,t']$ and $(t',\,T]$. We can use this circumstance first to decompose (13) exactly at the time instances where observations are present, so that we need to evolve a finite set of integral equations with purely convolutional kernel and individual initial conditions. Thus, given a time-mesh $t_0 < t_1 < \cdots < t_K$, where we set each $t_k$ exactly at the times of evidence, we obtain the following equation for the interpolant $\psi_k(x,\,t)$ at the $k$-th time interval $t \in (t_k,\,t_{k+1}]$

$$\psi_k(x,\,t) = \int_{t_k}^{t} \mathrm{d}\tau\, f(t-\tau \mid x)\, \upsilon_k(x,\,\tau)\, \phi_k(x,\,\tau) + \sum_{m=0}^{k-1} \int_{t_m}^{t_{m+1}} \mathrm{d}\tau\, f(t-\tau \mid x)\, \upsilon_k(x,\,\tau)\, \phi_m(x,\,\tau) + \upsilon_k(x,\,0)\, g_\psi(t)$$

with the current (normalized) likelihood update function $\upsilon_k(x,\,\tau) \equiv \upsilon(x,\,\tau,\,t)$ if and only if $t \in (t_k,\,t_{k+1}]$. As long as we are in this interval, $\upsilon(\tau,\,t,\,x)$ can be written using only the time-difference $\tau$ as argument, since then its scaling does not change and we can understood $\upsilon_k(x,\,\tau)$ as a simple staircase function. Thus, denoting the unit step function with $u(t)$, we recombine $\phi'(x,\,t) \equiv \sum_{m=0}^{k} (u(t-t_m) - u(t-t_{m+1}))\, \upsilon_k(x,\,t)\, \phi_m(x,\,t)$ and can then obtain the equation

$$\psi_k(x,\,t) = \int_{0}^{t} \mathrm{d}\tau\, f(t-\tau \mid x)\, \phi'(x,\,t) + g_\psi'(x,\,t)$$

with $g_\psi'(t) \equiv \upsilon_k(x,\,0)\, g_\psi(x,\,t)$ while in the case of 12, we can do the same and obtain the similar equation for the interpolant $\alpha_k(x,\,t)$,

$$\alpha_k(x,\,t) = \int_{0}^{t} \mathrm{d}\tau\, \Lambda(t-\tau \mid x)\, \phi'(x,\,t) + g_p'(x,\,t)$$

with $g_p'(t) \equiv \upsilon_k(x,\,0)\, g_p(x,\,t)$. Since we have now obtained the "standard form" of both equations resembling 12 and 12, we can reperform the derivation for (20) to obtain

$$\frac{\mathrm{d}}{\mathrm{d}t} \alpha_k(x,\,t) = \sum_{x' \neq x} m(x \mid x') \int_{0}^{t} \mathrm{d}\tau\, \kappa(x',\,t-\tau)\, \upsilon_k(x,\,\tau)\, \alpha'(x',\,\tau) \tag{21}$$

$$- m(x' \mid x) \int_{0}^{t} \mathrm{d}\tau\, \kappa_k(x,\,t-\tau)\, \upsilon_k(x,\,\tau)\, \alpha'(x,\,\tau) + g'(x,\,t)$$

$$= \sum_{x' \neq x} m(x \mid x') \left( \omega_k^{(k)}(x') \int_{t_k}^{t} \mathrm{d}\tau\, \kappa(x',\,t-\tau)\, \alpha_k(x',\,\tau) + \sum_{m=0}^{k-1} \omega_k^{(m)}(x') \int_{t_m}^{t_{m+1}} \mathrm{d}\tau\, \kappa(x',\,t-\tau)\, \alpha_m(x',\,\tau) \right)$$

$$- m(x' \mid x) \left( \omega_k^{(k)}(x) \int_{t_k}^{t} \mathrm{d}\tau\, \kappa(x,\,t-\tau)\, \alpha_k(x,\,\tau) + \sum_{m=0}^{k-1} \omega_k^{(m)}(x) \int_{t_m}^{t_{m+1}} \mathrm{d}\tau\, \kappa(x,\,t-\tau)\, \alpha_m(x,\,\tau) \right)$$

$$+ \upsilon_k(x,\,0)\, g(x,\,t)$$

valid only in the time window $t \in [t_k,\,t_{k+1})$. The constant weights $\omega_k^{(m)}(x)$ are obtained from recognizing that $\omega_k^{(m)}(x) \equiv \upsilon_k(x,\,\tau) = \text{const.}, \forall \tau \in [t_m,\,t_{m+1})$. Writing (21) more compactly in matrix-vector and operator notation can be done e.g. by

$$\frac{\mathrm{d}}{\mathrm{d}t}\alpha = (\mathsf{M} - \mathsf{I})\,\mathsf{K}_\alpha\alpha + \upsilon_0 g \qquad \Leftrightarrow \qquad \frac{\mathrm{d}}{\mathrm{d}t}\boldsymbol{\alpha}\,(t) = (\boldsymbol{M} - \boldsymbol{I})\int_0^t \mathrm{d}\tau\,\boldsymbol{D}_\upsilon\,(\tau,\,t)\,\boldsymbol{K}\,(t - \tau)\,\boldsymbol{\alpha}\,(\tau) + \boldsymbol{D}_\upsilon\,(0,\,t)\,\boldsymbol{g}\,(t)$$

where the integral operator $\mathsf{K}_\alpha$ acts on a suitable function $g$ by $\mathsf{K}_\alpha g \equiv \int_0^t \mathrm{d}\tau\,\upsilon\,(x,\,\tau,\,t)\,\kappa\,(x,\,t - \tau)\,g\,(x,\,\tau)$.

### A.3.3  Backward Equation under Observations

We now derive the backward equation on the marginal likelihoods $\beta\,(x,\,t)$ consistent with the formulations in this work. First, assume there is an end condition given but there are no intermediate observations. We mark this by introduction of the namimg $\beta_{\mathrm{no}}\,(x,\,t)$ for marginal likelihoods corresponding to this case. We will first replicate an equation similar to (20) for the backward case. The derivation is similar to that of the forward equation, so that it uses a combination of equations (16), (18) and (19) while leaving out the observation likelihood function. The combination is again carried out using the Laplace transform. For this, note that we can rewrite (16) such that

$$\phi'_\beta\,(x,\,t') = \int_0^{t'} \mathrm{d}\tau\,f\,(t' - \tau \mid x)\,\psi'_\beta\,(x,\,\tau) + \gamma'_\phi\,(x,\,t')$$

with the substitutes $\phi'_\beta\,(x,\,t') \equiv \phi_\beta\,(x,\,T - t')$, $\psi'_\beta\,(x,\,\tau) \equiv \psi_\beta\,(x,\,T - \tau)$ and $\gamma'_\phi\,(x,\,t') \equiv \gamma'_\phi\,(x,\,T - \tau)$. We can do the analogue with equation (18) to carry out similar steps to section A.3.1 and obtain

$$\frac{\mathrm{d}}{\mathrm{d}t}\beta_{\mathrm{no}}\,(x,\,t) = \sum_{x' \neq x} m\,(x \mid x')\int_t^T \mathrm{d}\tau\,\kappa\,(x',\,\tau - t)\,\beta_{\mathrm{no}}\,(x,\,\tau) - m\,(x' \mid x)\int_t^T \mathrm{d}\tau\,\kappa\,(x,\,\tau - t)\,\beta_{\mathrm{no}}\,(x',\,\tau) + \gamma\,(x,\,t) \quad (22)$$

with the memory kernel $\kappa\,(x,\,t) \equiv \mathsf{L}^{-1}\left(\frac{\tilde{f}(x,\,s)}{\tilde{\Lambda}(x,\,s)}\right)$ coinciding with the one of the forward equation. Matrix-vector notation again reveals additional structure, so that we can write

$$\frac{\mathrm{d}}{\mathrm{d}t}\boldsymbol{\beta}_{\mathrm{no}}\,(t) = \left(\boldsymbol{I} - \boldsymbol{M}^{\mathrm{T}}\right)\int_t^T \mathrm{d}\tau\,\boldsymbol{K}\,(\tau - t)\,\boldsymbol{\beta}_{\mathrm{no}}\,(\tau)$$

Observations can then be added similarly to section A.3.1. First, we apply again the time-reversal substitution like we did for (22) but this time including the substitution of the reversed normalized observation likelihood functions $\upsilon'_k\,(x,\,\tau) \equiv \upsilon\,(x,\,T - t,\,T - \tau)$ like we did to derive (21). Then, again strictly following the steps in section A.3.1 - which also led to (21) - and then reversing the substitutions, we obtain

$$\frac{\mathrm{d}}{\mathrm{d}t}\beta\,(x,\,t) = \sum_{x' \neq x} m\,(x \mid x')\int_t^T \mathrm{d}\tau\,\kappa\,(\tau - t)\,\upsilon'_k\,(x,\,\tau)\,\beta\,(x,\,\tau) - m\,(x' \mid x)\int_t^T \mathrm{d}\tau\,\kappa\,(\tau - t)\,\upsilon'_k\,(x',\,\tau)\,\beta\,(x',\,\tau) + \gamma\,(x,\,t)$$

$$(23)$$

$$= \sum_{x' \neq x} m\,(x \mid x')\left(\varpi_k^{(k)}\,(x)\int_t^{t_k} \mathrm{d}\tau\,\kappa\,(x,\,\tau - t)\,\beta_k\,(x,\,\tau) + \sum_{m=k}^{K-1} \varpi_k^{(m)}\,(x)\int_{t_m}^{t_{m+1}} \mathrm{d}\tau\,\kappa\,(x,\,\tau - t)\,\beta_m\,(x,\,\tau)\right)$$

$$+ m\,(x' \mid x)\left(\varpi_k^{(k)}\,(x')\int_t^{t_k} \mathrm{d}\tau\,\kappa\,(x',\,\tau - t)\,\beta_k\,(x',\,\tau) + \sum_{m=k}^{K-1} \varpi_k^{(m)}\,(x')\int_{t_m}^{t_{m+1}} \mathrm{d}\tau\,\kappa\,(x',\,\tau - t)\,\beta_m\,(x',\,\tau)\right)$$

valid only in the time window $t \in (t_k, t_{k+1}]$. The constant weights $\varpi_k^{(m)}(x)$ are again obtained from recognizing that $\varpi_k^{(m)} \equiv \upsilon_k'(x, \tau) = \text{const.}, \forall \tau \in [t_m, t_{m+1})$. Writing (21) more compactly in matrix-vector and operator notation can be done e.g. by

$$\frac{\mathrm{d}}{\mathrm{d}t}\beta = \left(\mathsf{I} - \mathsf{M}^\dagger\right) \mathsf{K}_\beta \beta + \upsilon_0 g \qquad \Leftrightarrow \qquad \frac{\mathrm{d}}{\mathrm{d}t}\boldsymbol{\beta}(t) = \left(\boldsymbol{I} - \boldsymbol{M}^\mathrm{T}\right) \int_t^T \mathrm{d}\tau \, \boldsymbol{D}_\upsilon(t, \tau) \boldsymbol{K}(\tau - t) \boldsymbol{\beta}(\tau) + \boldsymbol{D}_\upsilon(t, T) \boldsymbol{g}(t)$$

where the integral operator $\mathsf{K}_\beta$ acts on a suitable function $g$ by $\mathsf{K}_\beta g \equiv \int_t^T \mathrm{d}\tau \, \upsilon(x, \tau, t) \kappa(x, t - \tau) g(x, \tau)$ .

## A.4 Posterior/Smoothed Probabilities

An expression for the smoothed posterior marginals is available, since we can make use of the Markov property at the times of state transitions. Although we cannot simply multiply $\alpha(x, t) \equiv \lim_{h \to 0} \frac{1}{h} P\left(X(t) = x \mid \mathbf{y}_{[0, t)}\right)$ and $\beta(x, t) \equiv \lim_{h \to 0} P\left(\mathbf{y}_{[t, T)} \mid X(t) = x\right)$ like we could in a Markov chain. Nonetheless, on our way to derive the forward and backward equations we have used that $\tilde{p}(x, s) = \tilde{\Lambda}(x, s)\tilde{\phi}(x, s) + \tilde{g}_p(x, s)$ and $\tilde{\psi}(x, s) = \tilde{f}(x, s)\tilde{\phi}(x, s) + \tilde{g}_\phi(x, s)$ (rf. section A.3.1), from which we have seen that

$$\tilde{\psi}(x, s) = \frac{\tilde{f}(x, s)}{\tilde{\Lambda}(x, s)} \left(\tilde{p}(x, s) - \tilde{g}_p(x, s)\right) + \tilde{g}_\psi(x, s)$$

Under application of our boundary conditions and incorporation of the observations, we thus have for the forward and backward cases

$$\boldsymbol{\psi}_\alpha(t) = \int_0^t \mathrm{d}\tau \, \boldsymbol{D}_\upsilon(\tau, t) \boldsymbol{K}(t - \tau) \boldsymbol{\alpha}(\tau) + \boldsymbol{g}(t) \qquad \bigg| \qquad \boldsymbol{\phi}_\beta(t) = \int_t^T \mathrm{d}\tau \, \boldsymbol{D}_\upsilon(t, \tau) \boldsymbol{K}(\tau - t) \boldsymbol{\beta}(\tau) + \boldsymbol{\gamma}(t) \qquad (24)$$

We know from (3) that $\frac{\mathrm{d}}{\mathrm{d}t}p(x, t) = \sum_{x' \neq x} m(x \mid x') \psi(x', t) - m(x' \mid x) \psi(x, t)$ for the observation-less forward derivative. We also used a modified version for the forward currents and derived a similar relationship for the backward currents in (15). We thus assume, for the posterior still holds $\frac{\mathrm{d}}{\mathrm{d}t}\hat{p}(x, t) = \sum_{x' \neq x} m(x \mid x') \hat{\psi}(x', t) - m(x' \mid x) \hat{\psi}(x, t)$ where $\hat{\psi}(x, t)$ is the posterior output current. This should hold in between observations. At observation times, there can still be discontinuities. We also know from (15) that $\sum_{x' \neq x} m(x' \mid x) \phi_\beta(x', t) = \psi_\beta(x, t)$. Combining both

$$\begin{aligned}
\hat{\psi}(x, t) &= \psi_\alpha(x, t) \psi_\beta(x, t) \\
&= \lim_{h \to 0} \frac{P\left(\mathbf{y}_{(t, T]}, \mid X(t + h) \neq x, X(t) = x\right)}{P\left(\mathbf{y}_{(t, T]}, \mid \mathbf{y}_{(0, t]}\right)} \frac{1}{h} P\left(X(t + h) \neq x, X(t) = x \mid \mathbf{y}_{(0, t]}\right) \\
&= \lim_{h \to 0} \frac{1}{h} P\left(X(t + h) \neq x, X(t) = x, \mathbf{y}_{(0, T]}\right)
\end{aligned}$$

However, numerical errors from integrating the currents will accumulate in this approach and thus, another way is chosen to resolve the marginals. This is also similar to one used in HSMM's [3]. We get to it be considering the path we formulated in (17). Howevr, this time we are not assuming an uninformed past but insert the product of the normalized observation likelihood $\frac{P\left(\mathbf{y}_{[t-\tau, t)} \mid \mathbf{x}_{[t-\tau, t)} = x\right)}{P\left(\mathbf{y}_{[t-\tau, t)} \mid \mathbf{y}_{[0, t)}\right)}$ gathered in $[t - \tau, t)$ and the filtered current $P\left(X(t - \tau' + h) = x, X(t - \tau') \neq x \mid \mathbf{y}_{(0, t - \tau']}\right)$ instead. Then, we end up at a similar equation to (18) but with $\phi_\alpha$ and additional observations under the integral. Thus,

$$\hat{p}(x, t) = \int_0^\infty d\tau \int_0^\infty d\tau' \, \upsilon(x, t - \tau', t + \tau) \, \phi_\alpha(t - \tau') \, f(\tau' + \tau) \, \psi_\beta(x, t + \tau)$$

$$= \int_0^{T-t} d\tau \int_0^t d\tau' \, \upsilon(x, t - \tau', t + \tau) \, \phi_\alpha(t - \tau') \, f(\tau' + \tau) \, \psi_\beta(x, t + \tau)$$

$$+ \int_0^{T-t} d\tau \, \upsilon(x, 0, t + \tau) \, g_\phi(t + \tau) \, \psi_\beta(x, t + \tau)$$

$$+ \int_0^t d\tau' \, \upsilon(x, t - \tau', T) \, \phi_\alpha(t - \tau') \, \gamma_\psi(t - \tau') \tag{25}$$

which is the equation we used to determine the posterior marginals $\hat{p}$ from the calculated forward and backward currents $\phi_\alpha$ and $\psi_\beta$.

## A.5 Maximum A-Posteriori Estimation of Trajectories (Viterbi-type Algorithm)

The path marginalization in (5) - which expresses $\psi$ by all previous values of $\phi$ - quantifies infinitesimally how likely it is to exit state $x$ at time $t$ irrespective of the specific trajectory traversed before $t$. We can, however, also ask how likely it is to exit state $x$ at time $t$ with respect to the most likely previous trajectory before $t$. This rationale is the core of our Viterbi-type algorithm which - analogous to the original Viterbi algorithm - propagates a single per-state quantity. When the end of the time-window of consideration is reached, this per-state quantity is sufficient to backtrack the MAP trajectory. Necessary condition for this to work is that given these quantities, further evolution satisfies the Markov property. In the CTSMC, such quantities are the currents. Note, that the transition dictionary often additionally defined in the original Viterbi algorithm is completely optional and intended for speedup.

Let us first again take a closer look at the marginalized (unnormalized) input current $\check{\phi}_\alpha$. There, we have

$$\check{\phi}_\alpha(x, t) \equiv \lim_{h \to 0} \frac{1}{h} P\left(X(t + h) = x, X(t) \neq x, \mathbf{y}_{[0, t)}\right) = \int_{\mathbf{S}_{[0, t)}^x} \lim_{h \to 0} \frac{1}{h} P\left(X(t + h) = x, X(t) \neq x, \mathbf{x}_{[0, t)}, \mathbf{y}_{[0, t)}\right) d\mu_{x, t}$$

where $\mu_{x, t}$ is the probability measure of all partial trajectories $\mathbf{x}_{[0, t)}$ with support over the whole set of possible trajectories $\mathbf{S}_{[0, t)}^x$ from 0 to $t$ that end in a different state than $x$. Since Viterbi's original algorithm does only work on Markov chains with a fixed number of transitions, we need to keep track of the transition number $n$. We also call this number the "chain length". Instead of the marginalization, we demand a supremum giving the maximum input current $\phi_{\max}$, for which

$$\phi_{\max}(x, t; n) \equiv \sup_{\mathbf{x}_{[0, t)} \in \mathbf{S}_{[0, t)}^x} \lim_{h \to 0} \frac{1}{h} P\left(X(t + h) = x, X(t) \neq x, \mathbf{x}_{[0, t)}, \mathbf{y}_{[0, t)} \mid n\right)$$

This maximum current can be propagated in the following way. The current $\phi_{\max}(x, t)$ gives the infinitesimal probability of a state change to $x$ at $t$ of the most likely trajectory so far. A CTSMC has the Markov property at jump times. Thus, if we want to determine the most likely trajectory at a jump instance $t$, we need to look for the most likely total trajectory

$$\lim_{h \to 0} \frac{1}{h} P\left(X\left(t+h\right) = x,\ X\left(t\right) \neq x,\ \mathbf{x}_{[0,\,t)},\ \mathbf{y}_{[0,\,t)}\ \mid\ n+1\right) = \lim_{h \to 0} P\left(X\left(t+h\right) = x\ \mid\ X\left(t+h\right) \neq x,\ X\left(t\right) = x'\right)$$

$$\times \frac{1}{h} P\left(X\left(t+h\right) \neq x,\ X\left(t\right) = x',\ \mathbf{x}_{[t',\,t)} = x',\ \mathbf{y}_{[t',\,t)}\ \mid\ X\left(t'+h\right) = x',\right.$$

$$\times \frac{1}{h} P\left(X\left(t'+h\right) = x',\ X\left(t'\right) \neq x',\ \mathbf{x}_{[0,\,t')},\ \mathbf{y}_{[0,\,t')}\ \mid\ n\right)$$

for which we need the supremum. Thus, we obtain for the maximum input current $\phi_{\max}\left(x,\ t;\ n\right)$ the following identity

$$\phi_{\max}\left(x,\ t;\ n+1\right) = \sup_{x' \neq x,\, t' \in [0,\,t)} m\left(x\ \mid\ x'\right) \check{v}\left(x',\ t',\ t\right) f\left(t-t'\ \mid\ x'\right) \phi_{\max}\left(x',\ t';\ n\right) \tag{26}$$

Then, the most likely terminal state $x$ at $T$ can be found similarly by the analogoous equation to (7), i.e.

$$\sup_{x} P\left(\mathbf{x}_{[0,\,T)},\ \mathbf{y}_{[0,\,T)}\ \mid\ n\right) = \sup_{x,\, t' \in [0,\,T)} \lim_{h \to 0} \frac{1}{h} P\left(\mathbf{x}_{[t',\,t)} = x',\ \mathbf{y}_{[t',\,t)}\ \mid\ X\left(t'+h\right) = x',\ X\left(t'\right) \neq x\right)$$

$$\times \frac{1}{h} P\left(X\left(t'+h\right) = x',\ X\left(t'\right) \neq x',\ \mathbf{x}_{[0,\,t')},\ \mathbf{y}_{[0,\,t')}\right)$$

$$= \check{v}\left(x,\ t',\ t\right) \Lambda\left(t-t'\ \mid\ x\right) \phi_{\max}\left(x,\ t';\ n\right) \tag{27}$$

and thus

$$\sup_{x} P\left(\mathbf{x}_{[0,\,T)},\ \mathbf{y}_{[0,\,T)}\right) = \sup_{x,\,n} P\left(\mathbf{x}_{[0,\,T)},\ \mathbf{y}_{[0,\,T)}\ \mid\ n\right) P\left(n\right)$$

Note, that the chain length $n$ is important to track. Otherwise, we would ignore the differences in absolute scale of the infinitesimal probability of paths of different amounts of degrees of freedom. Imagine, we had all exponential waiting time distributions, so that for each $f\left(\tau\ \mid\ x\right)$ the mode lies at $\tau = 0$. Then, falsely, the most likely trajectory would have an infinitude of transitions while the total probability of such a jump count is going to zero. Thus, for these values to become comparable, we must resolve the conditional $n$ and scale them by the proper normalizer, which is the probability of ecountering so many transitions.

For an optimization problem like this, we don't need knowledge of the actual $P\left(n\right)$, we just need a value $L\left(n\right) \propto P\left(n\right)$. To determine such an $L\left(n\right)$, we can think of how we resolve the question if a random variable $Z_3$, which is a sum of two random variables $Z_1 + Z_2$, is larger than a certain value $T$. Additionally, the distributions of $Z_1$ and $Z_2$ are known and their respective probability densities $z_1\left(t\right)$ and $z_2\left(t\right)$ exist. To answer the question, we would then build the sum distribution, which is the convolution of $Z_1$ and $Z_2$ over their joint support and sample the survival function at $T$. Instead of evaluating the survival function of the two random variables, we could also convolve the probability density of one and the survival function of the other. Hereby it doesn't matter, which one provides the density and which the survival function. This circumstance allows us to efficiently determine the chain length scaling $L\left(n\right)$. We build a sequence of currents which are build from previous currents by convolution, each associated with a chain length $n$. The currents are concurrently convolved with the survival functions of the waiting time distributions to calculate the factors $L\left(n\right)$.

Thus, we follow with sequence update equations for chain-length currents

$$\phi_{n+1}\left(x,\,t\right)=\sum_{x'\neq x}m\left(x\mid x'\right)\int_{0}^{t}\mathrm{d}\tau\,\check{v}\left(x',\,\tau,\,t\right)f\left(t-\tau\mid x'\right)\phi_{n}\left(x',\,\tau\right) \tag{28}$$

$$p_{n}\left(x,\,T\right)=\int_{0}^{T}\mathrm{d}\tau\,\check{v}\left(x,\,\tau,\,T\right)\Lambda\left(t-\tau\mid x\right)\phi_{n}\left(x,\,\tau\right) \tag{29}$$

where the second equation determines the probability that a $n+1$-th transition would happen outside the considered time-window, i.e. the jump count would be $n$ if state $x$ is held at $T$. Thus, we obtain $L\left(n\right)=\sum_{x}p_{n}\left(x,\,T\right)$ by marginalization over all states $x$. Then, we find the most probable terminal state by

$$\left(x^{*},\,n^{*}\right)=\arg\sup_{x,\,n}P\left(\mathbf{x}_{[0,\,T)},\,\mathbf{y}_{[0,\,T)}\mid n\right)L\left(n\right) \tag{30}$$

from which we can backtrack to obtain the MAP trajectory in the time window $[0,\,T)$. The initial conditions for (28) and (29) if given initial $g_{\phi}$ are

$$p_{0}\left(x,\,T\right)=\check{v}\left(x,\,0,\,T\right)\int_{T}^{\infty}\mathrm{d}\tau\,g_{\phi}\left(\tau\mid x\right) \tag{31}$$

$$\phi_{1}\left(x,\,t\right)=\sum_{x'\neq x}m\left(x\mid x'\right)\check{v}\left(x',\,0,\,t\right)g_{\phi}\left(x',\,t\right) \tag{32}$$

For $\phi_{\max}$ for $t<0$, we need either a pre-specified function or a clear boundary condition. E.g. if we know when the last transition happened before $t=0$, we can simply set the new $t\leftarrow t-t'$ and $T\leftarrow T+t'$ and start from there with $\phi_{\max}\left(x,\,t;\,0\right)\equiv\delta\left(t\right)\sum m\left(x\mid x'\right)\bar{\lambda}\left(x'\right)\bar{p}\left(x'\right)=\delta\left(t\right)\bar{\lambda}\left(x\right)\bar{p}\left(x\right)$. Otherwise, if we assume a steady state before the considered time-window, we can choose $\phi_{\max}\left(x,\,t;\,0\right)\equiv u\left(-t\right)\bar{\lambda}\left(x\right)\bar{p}\left(x\right)$ with the unit step function $u$.

# B    Numerical Methods

## B.1    Calculation of Integral Equations

In this work, we throughout used methods which yield single integral equations to be solved in a time-window $[t_{0},\,t_{1}]$ which we assume to be Lipschitz. Their integration kernel within the specified time-window is purely convolutional. The equations are usually of the type

$$\boldsymbol{x}\left(t\right)=\boldsymbol{Q}_{I}\int_{t_{0}}^{t}\mathrm{d}\tau\,\boldsymbol{D}_{k}\left(t-\tau\right)\boldsymbol{x}\left(\tau\right)+\boldsymbol{Q}_{g}\boldsymbol{g}\left(t\right)$$

with the diagonal matrix of convolutional kernel functions $\boldsymbol{D}_{k}\left(t\right)\equiv\operatorname{diag}\left(\kappa_{1}\left(t\right),\,\ldots,\,\kappa_{d}\left(t\right)\right)$ where $d$ is the dimension of the equation (most times $d=\left|\mathcal{X}\right|$, the number of states of the CTSMC in consideration). Such integral equations of the second kind as encountered in our work are solved in the following way. With respect to the time difference, an amount of grid points $N$ is chosen as stated in the main text. Let's assume, we already have $n$ points corresponding to times $\boldsymbol{t}\equiv\left(t_{1},\,\ldots,\,t_{n}\right)$. When we write instead $\boldsymbol{x}_{n}\equiv\left(x_{1,n},\,\ldots,\,x_{d,n}\right)$, we mean the vector of $d$ entries corresponding to time $t_{n}$. The inhomogeneity $\boldsymbol{g}:\mathbb{R}_{\geq0}\rightarrow\mathbb{R}^{d}$ that can be sampled continuously is also given. Then, along the time points $\boldsymbol{t}$, the

pre-sampled convolutional kernel functions $\boldsymbol{k}_n$ are multiplied with the given points and the trapezoidal quadrature rule is used to approximate the integrals. Then

$$\left(\boldsymbol{I} - \frac{h}{2}\boldsymbol{Q}_I\boldsymbol{k}_0\right)\boldsymbol{x}_{n+1} = \sum_{m=1}^{n} h\boldsymbol{k}_{n+1-m}\boldsymbol{x}_m + \frac{h}{2}\boldsymbol{k}_{n+1}\boldsymbol{x}_0 + \boldsymbol{Q}_g\boldsymbol{g}\left(t_{n+1}\right)$$

can be solved by a linear equation system solver (and should not be solved using matrix inverses). Successive calculation can then calculate along the whole domain $[t_0, t_1)$. When we instead encounter an integro-differential equation, i.e.

$$\frac{\mathrm{d}}{\mathrm{d}t}\boldsymbol{x}\left(t\right) = \boldsymbol{Q}_I\int_{t_0}^{t_1}\mathrm{d}\tau\,\boldsymbol{D}_k\left(t - \tau\right)\boldsymbol{x}\left(\tau\right) + \boldsymbol{Q}_g\boldsymbol{g}\left(t\right)$$

we end up with $\boldsymbol{x}_{n+1} \approx \boldsymbol{x}_n + h\frac{\mathrm{d}}{\mathrm{d}t}\boldsymbol{x}_n$ at

$$\left(\boldsymbol{I} - \frac{h^2}{2}\boldsymbol{Q}_I\boldsymbol{k}_0\right)\frac{\mathrm{d}}{\mathrm{d}t}\boldsymbol{x}_n = \frac{h}{2}\boldsymbol{k}_0\boldsymbol{x}_n + \sum_{m=1}^{n} h\boldsymbol{k}_{n+1-m}\boldsymbol{x}_m + \frac{h}{2}\boldsymbol{k}_{n+1}\boldsymbol{x}_0 + \boldsymbol{Q}_g\boldsymbol{g}\left(t_{n+1}\right)$$

which then also is solved using a linear equation system solver. This is an implicit backwards Euler method applicable to stiff equations. Afterwards, the obtained points $\boldsymbol{x}_n$ are then interpolated using cubic splines along the domain $[t_0, t_1)$ for each dimension $m \leq d$. This allows us to sample them as a new inhomogeneity $\boldsymbol{g}'$ in a following equation. Using such consecutive steps allows us simple integration of forward and backward equations (21) and (23).

## B.2 Adaptive HSMM Implementation

Our adaptive discretization strategy consists of maintaining a time-dependent set of samples of approximate currents and their timestamps $\Phi_\alpha\left(t\right) \equiv \{(\phi_{\alpha,k}, t_k) \mid t_K - t_0 < \Delta_{\max}, t_0 < \cdots < t_K, k = 0, \ldots, K\}$. An analogue set is built for $\psi_\beta$. Then, by using a sample-based quadrature rule, we can calculate the next sample. In this proposal, we use the trapezoidal rule and interpolate the points $(\phi_{\alpha,k}, t_k)$ using affine combinations, i.e. $\frac{t - t_k}{t_{k+1} - t_k}\phi_\alpha\left(x', t_{k+1}\right) + \frac{t_{k+1} - t}{t_{k+1} - t_k}\phi_\alpha\left(x', t_k\right)$ for $t \in [t_k, t_{k+1})$. Assume at $t_K$, the next observation is at $t'$. We take the trapezoidal rule and an implicit Euler-type step of size $h' \equiv \min\{h, t' - t_K\}$ to always hit exact observation times. Thus, with $t_{K+1} \equiv t_K + h'$ there exists a set of current likelihood update values $\omega_K\left(x, k\right) \equiv \upsilon\left(x, t_k, t_K\right)$ and we get

$$\phi_\alpha\left(x, t_{K+1}\right) \approx \sum_{x' \neq x} m\left(x \mid x'\right)\sum_{k=0}^{K}\int_{t_k}^{t_{k+1}}\mathrm{d}\tau\,\frac{f\left(t_{K+1} - \tau \mid x'\right)}{t_{k+1} - t_k}\left[(\tau - t_k)\phi_\alpha\left(x', t_{k+1}\right) + (t_{k+1} - \tau)\phi_\alpha\left(x', t_k\right)\right]$$

$$= \sum_{x' \neq x} m\left(x \mid x'\right)\sum_{k=0}^{K}\frac{\phi_\alpha\left(x', t_{k+1}\right) - \phi_\alpha\left(x', t_k\right)}{t_{k+1} - t_k}\int_{t_k}^{t_{k+1}}\mathrm{d}\tau\,\tau f\left(t_{K+1} - \tau \mid x'\right)$$

$$- \frac{\phi_\alpha\left(x', t_{k+1}\right)t_k - \phi_\alpha\left(x', t_k\right)t_{k+1}}{t_{k+1} - t_k}\int_{t_k}^{t_{k+1}}\mathrm{d}\tau\,f\left(t_{K+1} - \tau \mid x'\right)$$

$$= \sum_{x' \neq x} m\left(x \mid x'\right)\sum_{k=0}^{K} -\frac{\phi_\alpha\left(x', t_{k+1}\right) - \phi_\alpha\left(x', t_k\right)}{t_{k+1} - t_k}\left[\int_{t_{K+1} - t_k}^{t_{K+1} - t_{k+1}}\mathrm{d}\tau\,t_{K+1}f\left(\tau \mid x'\right) - \int_{t_{K+1} - t_k}^{t_{K+1} - t_{k+1}}\mathrm{d}\tau\,\tau f\left(\tau \mid x'\right)\right]$$

$$+ \frac{\phi_\alpha\left(x', t_{k+1}\right)t_k - \phi_\alpha\left(x', t_k\right)t_{k+1}}{t_{k+1} - t_k}\int_{t_{K+1} - t_k}^{t_{K+1} - t_{k+1}}\mathrm{d}\tau\,f\left(\tau \mid x'\right)$$

where

$$\int_{t_K-t_k}^{t_K-t_{k+1}} \mathrm{d}\tau \, \tau f\left(\tau \mid x'\right) = \left[\tau F\left(\tau \mid x'\right)\right]_{t_K-t_k}^{t_K-t_{k+1}} - \int_{t_K-t_k}^{t_K-t_{k+1}} \mathrm{d}\tau \, F\left(\tau \mid x'\right)$$
$$= \left(t_K - t_{k+1}\right) F\left(t_K - t_{k+1} \mid x'\right) - \left(t_K - t_k\right) F\left(t_K - t_k \mid x'\right)$$
$$- I_F\left(t_K - t_{k+1} \mid x'\right) + I_F\left(t_K - t_k \mid x'\right)$$

Then, resolving all integrals and sorting gives

$$\phi_\alpha(x,\, t_{K+1}) \approx \sum_{x' \neq x} m(x \mid x') \sum_{k=0}^{K} -\frac{\phi_\alpha(x',\, t_{k+1}) - \phi_\alpha(x',\, t_k)}{t_{k+1} - t_k} \left[ t_{K+1} F(t_{K+1} - t_{k+1} \mid x') - t_{K+1} F(t_{K+1} - t_k \mid x') \right.$$

$$\left. - (t_{K+1} - t_{k+1}) F(t_{K+1} - t_{k+1} \mid x') + (t_K - t_k) F(t_{K+1} - t_k \mid x') + I_F(t_{K+1} - t_{k+1} \mid x') - I_F(t_{K+1} - t_k \mid x') \right]$$

$$+ \frac{\phi_\alpha(x',\, t_{k+1})\, t_k - \phi_\alpha(x',\, t_k)\, t_{k+1}}{t_{k+1} - t_k} \left[ F(t_{K+1} - t_{k+1} \mid x') - F(t_{K+1} - t_k \mid x') \right]$$

$$= \sum_{x' \neq x} m(x \mid x') \sum_{k=0}^{K} F(t_{K+1} - t_{k+1} \mid x') \left[ \frac{\phi_\alpha(x',\, t_{k+1})\, t_k - \phi_\alpha(x',\, t_k)\, t_{k+1}}{t_{k+1} - t_k} \right.$$

$$\left. - \frac{\phi_\alpha(x',\, t_{k+1}) - \phi_\alpha(x',\, t_k)}{t_{k+1} - t_k} (t_{K+1} - t_{K+1} + t_{k+1}) \right]$$

$$+ F(t_{K+1} - t_k \mid x') \left[ (t_{K+1} - t_{K+1} + t_k) \frac{\phi_\alpha(x',\, t_{k+1}) - \phi_\alpha(x',\, t_k)}{t_{k+1} - t_k} \right.$$

$$\left. - \frac{\phi_\alpha(x',\, t_{k+1})\, t_k - \phi_\alpha(x',\, t_k)\, t_{k+1}}{t_{k+1} - t_k} \right]$$

$$+ (I_F(t_{K+1} - t_k \mid x') - I_F(t_{K+1} - t_{k+1} \mid x')) \frac{\phi_\alpha(x',\, t_{k+1}) - \phi_\alpha(x',\, t_k)}{t_{k+1} - t_k}$$

$$= \sum_{x' \neq x} m(x \mid x') \sum_{k=0}^{K} F(t_{K+1} - t_{k+1} \mid x')$$

$$\times \left[ \frac{\phi_\alpha(x',\, t_{k+1})\, t_k - \phi_\alpha(x',\, t_k)\, t_{k+1} - \phi_\alpha(x',\, t_{k+1})\, t_{k+1} + \phi_\alpha(x',\, t_k)\, t_{k+1}}{t_{k+1} - t_k} \right]$$

$$+ F(t_{K+1} - t_k \mid x') \left[ \frac{\phi_\alpha(x',\, t_{k+1})\, t_k - \phi_\alpha(x',\, t_k)\, t_k - \phi_\alpha(x',\, t_{k+1})\, t_k + \phi_\alpha(x',\, t_k)\, t_{k+1}}{t_{k+1} - t_k} \right]$$

$$+ (I_F(t_{K+1} - t_k \mid x') - I_F(t_{K+1} - t_{k+1} \mid x')) \frac{\phi_\alpha(x',\, t_{k+1}) - \phi_\alpha(x',\, t_k)}{t_{k+1} - t_k}$$

$$= \sum_{x' \neq x} m(x \mid x') \sum_{k=0}^{K} F(t_{K+1} - t_{k+1} \mid x') \left[ \frac{\phi_\alpha(x',\, t_{k+1})\, (t_k - t_{k+1})}{t_{k+1} - t_k} \right]$$

$$+ F(t_{K+1} - t_k \mid x') \left[ \frac{\phi_\alpha(x',\, t_k)\, (t_{k+1} - t_k)}{t_{k+1} - t_k} \right]$$

$$+ (I_F(t_{K+1} - t_k \mid x') - I_F(t_{K+1} - t_{k+1} \mid x')) \frac{\phi_\alpha(x',\, t_{k+1}) - \phi_\alpha(x',\, t_k)}{t_{k+1} - t_k}$$

$$= \sum_{x' \neq x} m(x \mid x') \sum_{k=0}^{K} F(t_{K+1} - t_{k+1} \mid x') \left[ -\phi_\alpha(x',\, t_{k+1}) \right] + F(t_{K+1} - t_k \mid x') \left[ \phi_\alpha(x',\, t_k) \right]$$

$$+ \left( \frac{I_F(t_{K+1} - t_k \mid x') - I_F(t_{K+1} - t_{k+1} \mid x')}{t_{k+1} - t_k} \right) \left[ \phi_\alpha(x',\, t_{k+1}) - \phi_\alpha(x',\, t_k) \right]$$

$$= \sum_{x' \neq x} m(x \mid x') \sum_{k=0}^{K} \phi_\alpha(x',\, t_{k+1}) \left[ \frac{I_F(t_{K+1} - t_k \mid x') - I_F(t_{K+1} - t_{k+1} \mid x')}{t_{k+1} - t_k} - F(t_{K+1} - t_{k+1} \mid x') \right]$$

$$- \phi_\alpha(x',\, t_k) \left[ \frac{I_F(t_{K+1} - t_k \mid x') - I_F(t_{K+1} - t_{k+1} \mid x')}{t_{k+1} - t_k} - F(t_{K+1} - t_k \mid x') \right]$$

and by summing two consecutive steps, we get

$$\phi_\alpha\left(x',\,t_{k+2}\right)\left[\frac{I_F\left(t_{K+1}-t_{k+1}\mid x'\right)-I_F\left(t_{K+1}-t_{k+2}\mid x'\right)}{t_{k+2}-t_{k+1}}-F\left(t_{K+1}-t_{k+2}\mid x'\right)\right]$$

$$-\phi_\alpha\left(x',\,t_{k+1}\right)\left[\frac{I_F\left(t_{K+1}-t_{k+1}\mid x'\right)-I_F\left(t_{K+1}-t_{k+2}\mid x'\right)}{t_{k+2}-t_{k+1}}-F\left(t_{K+1}-t_{k+1}\mid x'\right)\right]$$

$$+\phi_\alpha\left(x',\,t_{k+1}\right)\left[\frac{I_F\left(t_{K+1}-t_{k}\mid x'\right)-I_F\left(t_{K+1}-t_{k+1}\mid x'\right)}{t_{k+1}-t_{k}}-F\left(t_{K+1}-t_{k+1}\mid x'\right)\right]$$

$$-\phi_\alpha\left(x',\,t_{k}\right)\left[\frac{I_F\left(t_{K+1}-t_{k}\mid x'\right)-I_F\left(t_{K+1}-t_{k+1}\mid x'\right)}{t_{k+1}-t_{k}}-F\left(t_{K+1}-t_{k}\mid x'\right)\right]$$

$$=[\ldots]+\phi_\alpha\left(x',\,t_{k+1}\right)\left[-\frac{I_F\left(t_{K+1}-t_{k+1}\mid x'\right)-I_F\left(t_{K+1}-t_{k+2}\mid x'\right)}{t_{k+2}-t_{k+1}}+F\left(t_{K+1}-t_{k+1}\mid x'\right)\right.$$

$$\left.+\frac{I_F\left(t_{K+1}-t_{k}\mid x'\right)-I_F\left(t_{K+1}-t_{k+1}\mid x'\right)}{t_{k+1}-t_{k}}-F\left(t_{K+1}-t_{k+1}\mid x'\right)\right]$$

$$=[\ldots]+\phi_\alpha\left(x',\,t_{k+1}\right)\left[\frac{I_F\left(t_{K+1}-t_{k}\mid x'\right)-I_F\left(t_{K+1}-t_{k+1}\mid x'\right)}{t_{k+1}-t_{k}}-\frac{I_F\left(t_{K+1}-t_{k+1}\mid x'\right)-I_F\left(t_{K+1}-t_{k+2}\mid x'\right)}{t_{k+2}-t_{k+1}}\right]$$

We introduce the factors

$$\Delta_I\left(t_{K+1},\,k,\,x'\right)\equiv\frac{I_F\left(t_{K+1}-t_{k}\mid x'\right)-I_F\left(t_{K+1}-t_{k+1}\mid x'\right)}{t_{k+1}-t_{k}}$$

And so

$$\phi_\alpha\left(x,\,t_{K+1}\right)\approx\sum_{x'\neq x}m\left(x\mid x'\right)\phi_\alpha\left(x',\,t_{K+1}\right)\left[\Delta_I\left(t_{K+1},\,K,\,x'\right)-F\left(0\mid x'\right)\right]-m\left(x\mid x'\right)\phi_\alpha\left(x',\,0\right)\left[\Delta_I\left(t_{K+1},\,0,\,x'\right)-F\left(t_{K+1}\mid x'\right)\right]$$

$$+m\left(x\mid x'\right)\sum_{k=1}^{K}\phi_\alpha\left(x',\,t_{k}\right)\left[\Delta_I\left(t_{K+1},\,k-1,\,x'\right)-\Delta_I\left(t_{K+1},\,k,\,x'\right)\right]$$

We put all $t_K$-dependent terms on the lhs

$$\phi_\alpha\left(x,\,t_{K+1}\right)-\sum_{x'\neq x}m\left(x\mid x'\right)\left[\Delta_I\left(t_{K+1},\,K,\,x'\right)-F\left(0\mid x'\right)\right]\phi_\alpha\left(x',\,t_{K+1}\right)$$

$$\approx\sum_{x'\neq x}m\left(x\mid x'\right)\sum_{k=1}^{K}\phi_\alpha\left(x',\,t_{k}\right)\left[\Delta_I\left(t_{K+1},\,k-1,\,x'\right)-\Delta_I\left(t_{K+1},\,k,\,x'\right)\right]+\phi_\alpha\left(x',\,0\right)\left[\Delta_I\left(t_{K+1},\,0,\,x'\right)-F\left(t_{K+1}\mid x'\right)\right]$$

Then, with introduction of

$$\boldsymbol{J}\left(t_{K+1},\,K\right)\equiv\operatorname{diag}\left(\left[\Delta_I\left(t_{K+1},\,K,\,x_1\right)\right],\,\ldots,\,\left[\Delta_I\left(t_{K+1},\,K,\,x_N\right)\right]\right)$$

$$\boldsymbol{F}\left(t_{K+1},\,t_{k}\right)\equiv\operatorname{diag}\left(F\left(t_{K+1}-t_{k}\mid x_1\right),\,\ldots,\,F\left(t_{K+1}-t_{k}\mid x_N\right)\right)$$

we get the final update rule for the forward current $\phi_\alpha (t_{K+1})$.

$$[\boldsymbol{I} - \boldsymbol{M} (\boldsymbol{J} (t_{K+1}, K) - \boldsymbol{F} (t_{K+1}, t_{K+1}))] \, \boldsymbol{\phi}_\alpha (t_{K+1}) \tag{33}$$
$$\approx \boldsymbol{M} \left( \sum_{k=1}^{K-1} [\boldsymbol{J} (t_{K+1}, k-1) - \boldsymbol{J} (t_{K+1}, k)] \, \boldsymbol{\phi}_\alpha (t_k) + [\boldsymbol{J} (t_{K+1}, 0) - \boldsymbol{F} (t_{K+1}, 0)] \, \boldsymbol{\phi}_\alpha (0) \right)$$

which should at this point be solved using a linear equation solver instead of isolation of $\boldsymbol{\phi}_\alpha (t_{K+1})$ using a matrix inverse. The solution to (33) is calculated two successive steps for $r = \frac{1}{2}$ to get $\phi_\alpha^{\left(\frac{1}{2}\right)} (x, t_{K+1})$ and one time with $r = 1$ to get $\phi_\alpha^{(1)} (x, t_{K+1})$. Then, for an error $E \equiv \left| \phi_\alpha^{(1)} (x, t_{K+1}) - \phi_\alpha^{\left(\frac{1}{2}\right)} (x, t_{K+1}) \right|$ the step size is updated by $h \to \min \left\{ \max \left\{ h \frac{\text{tol}}{E}, h_{\max} \right\}, h_{\min} \right\}$. After each step $\Phi_\alpha (t)$ is updated by throwing away each sample for which $t_{K+1} - t_k < \Delta_{\max}$ for a chosen $\Delta_{\max} \in \mathbb{R}_{\geq 0}$. The backwards analogue is then

$$\left[ \boldsymbol{I} - \boldsymbol{M}^\mathrm{T} (\boldsymbol{J} (t_{K+1}, K) - \boldsymbol{F} (t_{K+1}, t_{K+1})) \right] \boldsymbol{\psi}_\beta (T - t_{K+1})$$
$$\approx \boldsymbol{M}^\mathrm{T} \left( \sum_{k=1}^{K} [\boldsymbol{J} (t_{K+1}, k-1) - \boldsymbol{J} (t_{K+1}, k)] \, \boldsymbol{\psi}_\beta (T - t_k) + [\boldsymbol{J} (t_{K+1}, 0) - \boldsymbol{F} (t_{K+1}, 0)] \, \boldsymbol{\psi}_\beta (T) \right)$$

After $\Phi_\alpha (t)$ and $\Psi_\beta (t)$ have been sampled for $t \in [0, T]$, they are interpolated by splines (e.g. corresponding to the quadrature rule chosen) and quadrature is done on (25) to obtain the approximate values for the posterior marginals $\hat{p} (t)$. We can also just calculate the forward and backward values using quadrature on (12) and (18). Also, this derivation can without much effort be repeated on the extended Kolmogorov equations to obtain a self-contained evolution of forward or backward marginals in the adaptive step-size HSMM.

# C    Algorithms

## C.1    Unidirectional Forward and Backward Algorithms

For the unidirectional forward and backward equations, we use a finite-element approach, since it allows evaluation of (20) and (22) to be done by many more available solvers and methods which don't deal with discontinuities. The following two equation is valid for the forward calculation within a time-windows $[t_k, t_{k+1})$ between two observations:

$$\alpha_k = (\mathsf{M} - \mathsf{I}) \left[ \mathsf{K}_{[t_k, t)} \alpha_k + g_k \right] \qquad \Leftrightarrow \qquad \boldsymbol{\alpha}_k (t) = (\boldsymbol{M} - \boldsymbol{I}) \left[ \int_{t_k}^{t} \mathrm{d}\tau \, \boldsymbol{K} (t - \tau) \, \boldsymbol{\alpha}_k (\tau) + \boldsymbol{g}_k (t) \right] \tag{34}$$

where the integral operator $\mathsf{K}_{[a,b)}$ is given by $\mathsf{K}_{[a,b)} g \equiv \int_a^b \mathrm{d}\tau \, \kappa (x, t - \tau) \, g (x, \tau)$. Similar to the forward case, we need the analogue equation for the backward case. This is given by

$$\beta_k = \left( \mathsf{I} - \mathsf{M}^\dagger \right) \left[ \mathsf{K}_{(t, t_{k+1}]} \beta_k + \gamma_k \right] \qquad \Leftrightarrow \qquad \boldsymbol{\beta}_k (t) = \left( \boldsymbol{I} - \boldsymbol{M}^\mathrm{T} \right) \left[ \int_{t}^{t_{k+1}} \mathrm{d}\tau \, \boldsymbol{K} (\tau - t) \, \boldsymbol{\beta}_k (\tau) + \boldsymbol{\gamma}_k (t) \right] \tag{35}$$

where the integral operator $\mathsf{K}_{(a,b]}$ is given by $\mathsf{K}_{(a,b]} g \equiv \int_a^b \mathrm{d}\tau \, \kappa (x, t - \tau) \, g (x, \tau)$. The inhomogeneities $g_k (x, t)$ and $\gamma_k (x, t)$ are built online during runtime from previously calculated time-windows. This is explained in Algs. C.1 and C.1.

---

**Algorithm 1** Forward Algorithm

---

**Input:** Interval $[0, T)$; $K$ observations $\mathcal{E}$; initial $g_0(t)$
**Output:** $\alpha(t)$ as sequence of interpolants $\alpha_k(t)$, $k \in \{1, \ldots, K\}$
 1: Set $t_0 \equiv 0$, $t_{K+1} \equiv T$, $\omega \equiv 1$
 2: **for** $k \in \{0, \ldots, K\}$ and $x \in \mathcal{X}$ **do**
 3:     $\alpha_k \leftarrow \mathtt{Solve}$ (eq. (34), $t \in [t_k, t_{k+1})$)
 4:     $\omega \leftarrow \upsilon(x, t_k, t_{k+1})$
 5:     $g_{k+1} \leftarrow \omega \left[ \mathsf{K}_{[t_k, t_{k+1})} \alpha_k + g_k \right]$
 6: **end for**

---

 

---

**Algorithm 2** Backward Algorithm

---

**Input:** Interval $(0, T]$; $K$ observations $\mathcal{E}$; initial $\gamma_T(t)$
**Output:** $\beta(t)$ as sequence of interpolants $\beta_k(t)$, $k \in \{1, \ldots, K\}$
 1: Set $t_0 \equiv 0$, $t_{K+1} \equiv T$, $\gamma_{K+1} \equiv \gamma_T$, $\omega \equiv 1$
 2: **for** $k \in \{K+1, \ldots, 1\}$ and $x \in \mathcal{X}$ **do**
 3:     $\beta_k \leftarrow \mathtt{Solve}$ (eq. (35), $t \in [t_{k+1}, t_{k+1})$)
 4:     $\omega \leftarrow \upsilon(x, t_k, t_{k+1})$
 5:     $\gamma_{k-1} \leftarrow \omega \left[ \mathsf{K}_{(t_{k-1}, t_k]} \beta_k + \gamma_k \right]$
 6: **end for**

---

## C.2   Forward-Backward Algorithm

For the forward-backward algorithm, we need equations (24) in a finite-element reformulation similar to (34) and (35) from the unidirectional algorithms.

$$\psi_\alpha = \mathsf{K}_{[t_k, t)} \alpha_k + g_k \qquad \Bigg| \qquad \phi_\beta = \mathsf{K}_{(t, t_k]} \beta_k + \gamma_k$$

Note, that they can simply be calculated during the forward and backward passes by storing the result before composition with $(\mathsf{M} - \mathsf{I})$ and $(\mathsf{I} - \mathsf{M}^\dagger)$. After that, we calculate equation (25) to obtain $\hat{p}$. Equation (25) is not an integral equation of the second kind but a simple two.dimensional integral. We thus don't detail its calculation any further. The forward-backward algorithm is then found in Alg. C.2.

---

**Algorithm 3** Forward-Backward Algorithm

---

**Input:** Interval $[0, T]$; $K$ observations $\mathcal{E}$; Initial $g_0(t)$, $\gamma_T(t)$
**Output:** $\hat{p}(t)$ for $t \in [0, T]$
 1: $\psi_\alpha \leftarrow \mathtt{Execute}$ Alg. C.1
 2: $\phi_\beta \leftarrow \mathtt{Execute}$ Alg. C.1
 3: $\phi_\alpha \leftarrow \mathsf{M} \psi_\alpha$
 4: $\psi_\beta \leftarrow \mathsf{M}^\dagger \phi_\beta$
 5: $\hat{p} \leftarrow \mathtt{Solve}$ (25), $t \in [0, T]$)

---

## C.3   Viterbi Algorithm

The Viterbi algorithm is executed according to the details from section A.5. For this, we pre-calculate the chain length scaling factor $L(n)$ using a sequence of calculations of (28) and successive marginalizations $L(n) = \sum_x p_n(x, T)$ up

to a specified $n_{\max}$. Then, up to this $n_{\max}$, using equations (26) and (27), we calculate the (unnormalized) terminal state probability given chain length $P\left(\mathbf{x}_{[0,\,T)},\,\mathbf{y}_{[0,\,T)} \mid n\right)$. Using (30), we then determine the optimal terminal state $x^*$ and chain length $n^*$. From hereon, we backtrack using first (27) and then (26) decreasing $n^*$ until zero and saving the intermediate transition times and states in two vectors $\boldsymbol{t} \equiv (t_0,\,t_1,\,\ldots,\,t_{n_{\max}})$ and $\boldsymbol{x} \equiv (x_0,\,x_1,\,\ldots,\,x_{n_{\max}})$. Then, we have obtained the MAP trajectory within the range of allowed jump counts or chain lengths $n_{\max}$. The algorithm is given in Alg. C.3.

---

**Algorithm 4** Viterbi Algorithm

---

**Input:** Interval $[0, T)$; $K$ observations $\mathcal{E}$; initial $g(t)$; $n_{\max}$; initial $\phi_{max}(\cdot, \cdot; 0)$

**Output:** $n$, $\boldsymbol{t}$, $\boldsymbol{x}$

  1: $\phi_1 \leftarrow$ `Solve` (eq. (32), $g$)
  2: $\phi_{\max}(\cdot, \cdot; 1) \leftarrow$ `Solve` (eq. (26), $\phi_{\max}(\cdot, \cdot; 0)$)
  3: $L(0) \leftarrow$ `Solve` (eq. (31), $g$)
  4: **for** $k \in \{1, \ldots, n_{\max}\}$ **do**
  5:     $\phi_{k+1} \leftarrow$ (eq. (28), $\phi_k$)
  6:     $\phi_{\max}(\cdot, \cdot; k+1) \leftarrow$ (eq. (26), $\phi_{\max}(\cdot, \cdot; k)$)
  7:     $L(k) \leftarrow$ `Solve` (eq. (29), $\phi_k$)
  8: **end for**
  9: $(n, t_n, x_n) \leftarrow$ `Solve` (eq. (30), $L$, $\phi_{\max}$)
 10: $c \leftarrow n$
 11: **while  do** $c > 0$ $c \leftarrow c - 1$
 12:     $t_c, x_c \leftarrow$ `Arg` (eq. (26), $phi_{\max}(\cdot, \cdot; c+1)$)
 13: **end while**
 14: $\boldsymbol{t} \leftarrow (t_0, \ldots, t_n)$
 15: $\boldsymbol{x} \leftarrow (x_0, \ldots, x_n)$

---

Note, we can also obtain $n_{\max}$ adaptively. Let e.g. $m_L(n) \equiv \max_{k \leq n} L(k)$ be the maximum $L$ up to a current $n$. Then, we could simply detect if we already passed a highly probable region of possible chain lengths by breaking loop if tol $> \frac{L(n)}{m_L(n)}$ for some specified relative tolerance tol. If there are more than one highly probable regions of chain lengths separated by a lowly probable region, this detection won't work reliably. However, in most cases, this procedure should either suffice as is or can be adapted to check the condition for specifc period lengths depending on the structure imposed by the embedded Markov chain.

# D   Simulations

For improved accessibility (folders, separate files), further plots are moved to the repository:
    https://anonymous.4open.science/r/BFghJAGII31

## D.1   Numerical Calculation of Integral Equations

Erroneously, in the main text, we refer to section D for the calculation details of the integral equations. These are instead found in section B.1.

# E    Experiments

For improved accessibility (folders, separate files), documents specifying the experimental setups to the specific folders in the repository:

https://anonymous.4open.science/r/BFghJAGII31