# OpenReview forum: "Forward-Backward Latent State Inference for Hidden Continuous-Time semi-Markov Chains"
_NeurIPS.cc/2022/Conference — NeurIPS 2022 Accept_

### Official Review · Reviewer_6Ntb · 2022-07-10

**Rating:** 7
**Confidence:** 2
**Soundness:** 3 good
**Presentation:** 2 fair
**Contribution:** 3 good

**Summary:**

The paper generalizes latent state inference from HSMMs to continuous-time chains in latent space. In this case, the posterior is not simply proportional to the usual forward and backward probabilities. Instead, the transition random variables ("currents") are Markov. The authors take the limit of step size approaching 0 for these currents. The paper shows that the marginal $p(x, t)$, in this case, is found by a convolution over the input currents. Similarly, the output currents are a convolution over the input currents. Including observations, the authors derive Kolmogorov forward and backward equations, leading to algorithms akin to the classical sum-product (belief propagation) and max-sum (Viterbi) message-passing algorithms, but in continuous time. An additional advantage of the continuous formulation is the availability of adaptive step-sizing in the continuous case. The algorithms, as well as adaptive step-sizing, are evaluated in 1D experiments.

**Questions:**

- What is the black line in figures 2b,c? Is it the underlying ground truth state? If so, why does the blue HSMM with adaptive step size line in 2c seem to perform so poorly?

**Limitations:**

The authors properly discuss the limitations in the conclusion.

**Strengths And Weaknesses:**

Strengths:
- Strong theoretical foundation.
- Important generalization of well-known important algorithms to the continuous-time case.
- Adaptive step-sizing.

Weaknesses:
- The paper was a bit hard to follow from time to time. This could either be due to my lack of background knowledge but also perhaps due to the quite liberal suppression of function arguments and sometimes rather cluttered mathematics mid-sentence.


Minor:
- Some typos: l271: "therefor", l317: "differemces", l189: equation reference failed, l274: what does "(x)" refer to?

---

> ### Author Response · Authors · 2022-08-02
> **Initial response / accessibility / figure 2**
>
> We thank the reviewer for their careful review of our work. We tried to correct mentioned typos and grammatical errors and to conform to minor comments in a minor revision of the document issued for this rebuttal. In this revision, we also reworked our figures to include proper legends and axis descriptions. In the following we want to respond to major comments and questions which came up during review.
>
> In the minor revision, we reworked the equations a little to include more detail to make the connections between the steps clearer and improve their recognition value. We will also improve clarity in section 2.2 to make the setup needed for section 3 and 4 more accessible in the next major revision of the text.
>
> The black lines in figures 2b/c and also the left plot in 2d indeed show a representation of the latent trajectory $X\left(t\right)$. In the evaluation, the observations in $\mathcal{Y}\equiv\mathbb{R}$ are generated by the distribution $Y\left(t\right)\,|\,X\left(t\right)=x\,\sim\mathcal{N}\left(b\left(x\right),\,\sigma\right)$ with $\mathcal{N}$ being a normal distribution with standard deviation $\sigma$ and mean $b\left(x\right)$ with a function $b\,:\,\mathcal{X}\rightarrow\mathbb{R}$. This setup was chosen to be able to show meaningful continuous quantities (mean, variance) in the example scenarios. To visualize the ground truth latent trajectory, we show the result of the function $b$, i.e. the mean of $Y\left(t\right)$, at any time $t$ and chose $b$ to be invertible. The crosses in the figures show the final discretely sampled and noisy observations. Note, that the choice of $\mathcal{Y}$ is arbitrary and our algorithm only needs to be able to assign a likelihood to any observable event. Figure 2c shows an excerpt of the mean and variance of the estimated latent state representation, i.e. $b\left(X\left(t\right)\right)$, during a forward pass of an adaptive step HSMM (orange dots with error bars) and compares these quantities with mean and variance obtained from the continuous model (continuous blue line and blueish shaded region). The adaptive step HSMM in the scenario generates a relative linear interpolation error of below $10^{-3}$ w.r.t. to the fine-grained model as demanded by the hyperparameters. We chose a region of the pass with a single observation (cross) to show distinctly how the step-size changes overall and visibly increases with the chain approaching the steady state. Reducing computation in steady-state regions and adapting to the dynamic range of the process is an important justification for the adaptive step HSMM. However, we understand that this may give an undesired impression and will thus exchange figure 2c in the next revision to contain more than one observations. We also reworked section 6 to more clearly explain the evaluation setup.

---

> > ### Comment · Reviewer_6Ntb · 2022-08-06
> >
> > Thank you for the clarifications. I have no further questions.

---

### Official Review · Reviewer_if4n · 2022-07-11

**Rating:** 5
**Confidence:** 3
**Soundness:** 2 fair
**Presentation:** 2 fair
**Contribution:** 3 good

**Summary:**

This paper propose inference algortihms for hidden continuous time semi-Markov
chains (latent CTSMCs). Specifically it focuses on homogeneous and direction-time
independent CTSMCs. Both inference of posterior probabilities of latent states
given observations, as well as maximum-a posteriori state path are addressed.
The foundation of the algorithm is the observation that given a transition
event at a specific time, past and future dynamics become independant. Therefore,
the core element is to update the intensity of such event (termed current) as a
function of time and state. These updates involve integral equations that are solved
using numerical methods.


**Questions:**

**Suggestions**
1. The role of current is introduced in Section 2.1. At this point it will be
good to state what are the challenges in computing them and to outline the approach
for doing so.
2. Give detailed explanations about the initial conditions introduced after
Equation 5 and in the algorithm summary.
3. Give a short intuitive explanation about how integral equations are solved,
what is the complexity of the solvers and the overall method. This is important
in addition to the references that appear.
4. Provide a summary of all the steps involved in the algorithm.

**Questions and minor comments**
1. Lines 112, 114: Should 'leaving' be replaced by 'entering' in line 112 and vice
	versa in line 114?
2. Line 126: the current seem to be a density of a joint probability not a probability;
is likelihood is a better term?
3. Figure 1: Do the Gamma distributions refer to sojourn time?
4. Line 134: What is the meaning of 'assume a terminal condition to be fixed at $T$?
Does it mean the we observed the end state? Do we observe also how the entry time
to this state?
5. Line 141: the opening sentence is not clear.
6. **Equation 5:** what is $g_p$? Why is there an addition here? What are the properties
of this function? Also in line 153: What properties should $g_\phi$ satisfy?
7. Line 154: the motivation to demanding a transition shortly after $t$ is not
clear.
8. Line 189: fix the reference.
9. 201: Can you give a time-domain expression for $\kappa$ and explain what is gained
10. Equation 13: can you give an explicit reminder-reference there to how $\phi_\alpha$
and $\psi_\beta$ are computed as by-products?
11. Line 317: fix 'differemces'


**Limitations:**

The authors adequately addressed the limitations.

**Strengths And Weaknesses:**

Strengths:
This work tackles a very fundamental and important problem. The authors seem
exploit the structured representation of direction-time Independence in a elegant way,
namely by focusing on the representation of currents.

Weaknesses:
Although the setup is well written, starting from Section 2.2 the derivations become
hard to follow and it is hard to judge all the elements. The initial conditions are not
clear. Additionally, the numerical algorithm as well as the overall complexity is not
well described. See suggestions below regarding missing descriptions and flow.

---

> ### Author Response · Authors · 2022-08-02
> **Initial response / challenges in current calculation / initial conditions / solvers / algorithm in main text**
>
> We thank the reviewer for their careful review of our work. We tried to correct mentioned typos and grammatical errors and to conform to minor comments in a minor revision of the document issued for this rebuttal. In this revision, we also reworked our figures to include proper legends and axis descriptions. In the following we want to respond to major comments and questions which came up during review.
>
> 1. We agree with the assessment that after finishing section 2.1. the reader would benefit from more orientation regarding the next steps. We thus changed the title for the next section to a more meaningful one, i.e. "Obtaining the Forward and Backward Currents through Integral Equations". This change is included in the minor revision issued for this rebuttal.
>
> 2. We added a reference to section 4 where initial and terminal conditions are discussed at the first position where an initial condition ($g_{p}$) is mentioned. We also changed the title of the respective part of this section from "Boundary Conditions" to "Initial and Terminal Conditions" to make it more meaningful and improve its visibility in that regard.
>
> 3. In section 6, we have specified how the integral equations have been solved and there is a dedicated section in the supplement specifying the backwards Euler method used. While there exist variable step-size methods that make clever use of analytic properties, we didn't intend to put a too strong focus on numerics. We relied on the simple backwards Euler method for the evaluation to make sure the calculation is traceable, the results can be believed, and to minimize possible sources of errors from our side. Continuous equations that allow for variable-step solvers are generally preferable compared to a fixed uniform discretization. A general algorithmic time complexity (in the form of a worst-case runtime) for any possible solvers is hard to give, since it depends on the number of points chosen and/or the desired error bounds. We agree with the assessment that a remark on this numerical aspect may be interesting, but we think it is best placed in the supplement.
>
> As a result of content selection for the main text, we decided to leave the step-by-step description of the forward-backward and Viterbi algorithms to the supplementary material as referenced in the dedicated sections. We understand that such a description as part of the main text also helps to put the content into place. In a next major revision of the main text, we will consider an inclusion of the algorithmic descriptions.
>
> Questions:
>
> 1., 3., 8., 11. Fixed.
>
> 2. We reformulated to emphasize better on "per unit time".
>
> 3. We added "waiting time distributions".
>
> 4. The terminal condition is the backwards analogue to the initial condition and constitutes the event for which the likelihood is built. Having a final observation is probably not sufficient in a semi-Markov context, since the likelihood of that observation also depends on the infinite progression of the trajectory past the observation time instance. Like written in section 4, mainly two different conditions are typical which either fix an uninformed future or a transition after the final observation. However, depending on the application, other conditions can be reasonable.
>
> 5. We reformulated the sentence.
>
> 6. In line 153, there was a typo. $g_{\phi}$ has been exchanged by $g_p$. Otherwise, we see no ambiguities here. The introduction of $g_p$ implements the initial condition. The marginalization in (5) must be performed for all trajectories in the time-window $(-\infty,\,t)$ that lead to a residence in the considered state at time $t$. Since the path marginalization can be executed as a convolutional integral over the input currents, we gather the part integrating along $(-\infty,\,0)$ in the initial condition. Different initial conditions, i.e. assumptions on the history of the process, lead to different $g_p$'s. The reference to section 4 and to Appendix A is highly recommended to view the calculation of two sample cases. The procedure is a continuous-time analogue to what is done in discrete time (c.f. Yu's book from 2016).
>
> 7. Motivation has been made more clear.
>
> 9. A closed form expression for $\kappa$ in the time domain does not exist for a general choice of waiting time distributions. As written at the beginning of section 4, while the inference can also be done without the self-contained evolution equations for the forward and backward terms, such equations are generally of interest to the semi-Markov community and sometimes useful to find approximations or analyze memory properties.
>
> 10. Being able to compute $\phi_{\alpha}$ and $\psi_{\beta}$ as by-products arises as part of the derivation of the self-contained integro-differential forward and backward equations, now (10), in the older revision (10) and (11). We understand that this might be a bit obstructed and added a reference to Appendix A.

---

> > ### Comment · Reviewer_if4n · 2022-08-04
> > **Many clarity issues have been addressed**
> >
> > Thank you for clarifying a large part of the issues raised. I believe this will improve the readability of the manuscript and raising my rating from 4 to 5 to reflect that.

---

### Official Review · Reviewer_cruP · 2022-07-11

**Rating:** 6
**Confidence:** 3
**Soundness:** 3 good
**Presentation:** 2 fair
**Contribution:** 3 good

**Summary:**

The paper describes a class of models which the authors characterize as being "continuous time hidden semi-Markov chains" and provides an inferential framework for this class of models generalising the classical algorithms for HMM inference to this context.

**Questions:**

1. My first thought on seeing this paper was that in continuous time something like a hidden semi-Markov model is actually very much more natural than it is in discrete time where one has to jump through artificial hoops to induce the non-geometric holding time distribution. To what extent are hidden semi-Markov models different to the setting in which one has a latent piecewise deterministic Markov process in the sense of [Davis, M. H. A. (1984). "Piecewise-Deterministic Markov Processes: A General Class of Non-Diffusion Stochastic Models". Journal of the Royal Statistical Society. Series B (Methodological). 46 (3): 353–388. doi:10.1111/j.2517-6161.1984.tb01308.x. JSTOR 2345677.]?

2. Following up on 1, how does what is presented here relate to what is known for filtering in the setting in which one has a latent PDMP? As the PDMP class is much broader than that considered here it's very probable that substantial benefits can be obtained by considering this particular context but it would be nice to see this discussed a little.

3. Are there settings in which you envisage the method having real world application? The numerical example in this paper is underwhelming in that regard.

Minor details:
line 80: This seems to describe an overly general observation space Y given it's assumed to be real and to have a known conditional distribution function.
line 85: this is the intersection isn't what's intended, is it? It's the intersection of possibly several points and hence typically empty. I think what's wanted is actually the intesection of the events so the intersection hould come outside the { } and it might be clearer if the implicit \omega were made explicit.
(2) could be made clearer: what precisely does "Markov at jump instances" mean. Can you give a reference?
l189 (??)
l317 time-differemces,
l331 "detailled"
l342 10^-4 and 10^-3
l344 "adative"
l347 testet"


**Limitations:**

The authors have indicated that societal implications are N/A for this work and given the nature of the paper I think that is appropriate. I didn't note any particular limitations that are not discussed.

**Strengths And Weaknesses:**

This is a timely piece of work in the sense that there have been several innovations relating to inference for HSMMs in recent years and the development herein seems somewhat more principled than many of them. Working directly in continuous time seems to allow for a relatively clean derivation of forward and backward equations that facilitate the inferential tasks of interest. My impression is that there is currently interest in performing inference for this type of model and hence the work is likely to find an audience.

The main weaknesses, to my mind are:
1. The bulk of the paper is dedicated to fairly routine computations and the extent of the novelty is perhaps a little limited.
2. A lack of connection with the wider literature on hidden continuous time processes.
3. The empirical study lacks any compelling application or use case to showcase the model or demonstrate its broad importance.
4. The manuscript seems to have been a little hastily prepared and would benefit from careful proofreading (see the minor details in the next section for a few example typos)

---

> ### Author Response · Authors · 2022-08-02
> **Initial response / latent CTSMCs and PDMPs, POMPs / special properties / applications**
>
> We thank the reviewer for their careful review of our work. We tried to correct mentioned typos and grammatical errors and to conform to minor comments in a minor revision of the document issued for this rebuttal. In this revision, we also reworked our figures to include proper legends and axis descriptions. In the following we want to respond to major comments and questions which came up during review.
>
> 1. Discrete event data in continuous time assumed to be generated by an underlying discrete-state jump process is of course a very general generative model. The special case involving a latent semi-Markov chain has seen broad application in discrete time also because the generalization compared to a latent Markov chain is straight forward. Generalized models are available in the realm of not only PDMPs but also partially observable Markov processes (POMPs), where an unobservable dependent waiting-time process generates the observable semi-Markovian behaviour while the joint process is Markov. Representations of CTSMCs as members of each of the two process classes are available in the literature (PDMP: Book Cocozza-Thivent 2021; POMP: Book Limnios 2001). For POMPs as well as PDMPs - to our knowledge - we usually need an auxiliary memory variable $T\left(t\right)$ keeping track of the "current" waiting time (also called backward recurrence time in a Markov renewal context). The resulting process $\{X(t),T(t)\}$ is then a Markov process on the semi-continuous product state space $\mathcal{X}\times\mathbb{R}_{\geq0}$ (proof in Limnios 2001). In the case of a PDMP, we define a process $\{Z(t)\}$ associated with the flow $\varphi(X(t),T(t))$ (c.f. how Davis modelled the M/G/1 queue). In the POMP case, we work directly on $\{X(t),T(t)\}$ and define any observations to be only dependent on $X\left(t\right)$. Building the Kolmogorov equations (which take the form of PDEs) for $\{X(t),T(t)\}$ and calculating the integral equations along the boundary $T\left(t\right)=0$ results in the same equations we derived. While this had been a viable alternative approach, choosing the path marginalization instead was beneficial for us in two aspects. First, introduction of observation likelihood seemed to be most intuitive this way since we pictured constant partial trajectories, so it is clear how the likelihood function $P\left(Y\left(t\right)=y\,|\,X\left(t\right)=x\right)$ needs to be evaluated. Second, we could circumvent the technicality of introducing an explicit waiting time variable $T\left(t\right)$. We agree with the assessment that this context is of interest to some readers (also w.r.t. future research) and we will include a section in the supplement highlighting the context with a short reference in the main text.
>
> 2. The simple equations arise as a direct consequence of the renewal property of the CTSMC. Any derivation of the Kolmogorov equations we know is based on this property (c.f. Feller 1964, c.f. Gillespie 1977). While inference based on the latent Markov process $\{X(t),T(t)\}$ is in principle possible as well, Kolmogorov's equations for CTSMCs are usually considered as integro-differential equations in the component $X\left(t\right)$ alone. Besides again avoiding the introduction of a helper variable, the practical benefits of using this formulation match those of using boundary integral equations (BIEs) in boundary value problems described by PDEs. These usually don't reduce the dimension of the problem (here, looking at an semi-infinite history each time vs. looking at a semi-infinite density of $T\left(t\right)$) but allow numerical algorithms to exploit analytic properties of a 1D curve and/or allow efficient approximations of the involved (memory) integral. As the main exploitation of the CTSMC's properties, we might thus understand that the involved integral equations exist and exhibit simple expressions directly involving basic quantities of the waiting times. Because of their simple form covering constant partial trajectories in $X\left(t\right)$, we can show how these are modified when observation likelihood is included, how they can be used to derive two inference algorithms of sum-product and max-product type, and how they can be adaptively discretized to obtain a variable-step discrete model. As we write in section 4, the modified Kolmogorov equations arising are in principle not necessary for the marginal inference, but the availability of a self-contained evolution of forward and backward terms seemed to be important to communicate.
>
> 3. With respect to possible applications, we kindly refer to the respective part of the response we have also given to reviewer LD4m because of the character limit.
>
> -- sorry, this comment could not display the math we entered correctly. We hope, it is sufficiently readable

---

> > ### Comment · Reviewer_cruP · 2022-08-07
> > **Response**
> >
> > Thank you for the detailed response, which has clarified some things for me as a non-expert on this type of model. I would be willing to raise my rating from 5 to 6 as a result.

---

> > > ### Author Response · Authors · 2022-08-08
> > >
> > > We thank you for your assessment. Maybe it is an error on our side, but we cannot see the changed score in the author console. If what we can see is correct, we kindly ask you to adjust the score as advertised.

---

### Official Review · Reviewer_LD4m · 2022-07-15

**Rating:** 7
**Confidence:** 4
**Soundness:** 4 excellent
**Presentation:** 3 good
**Contribution:** 3 good

**Summary:**

This paper considers inference in Hidden Continuous-Time semi-Markov Chains. The authors derive a tractable algorithm for inference in these models. Notably, the authors consider the challenging case of irregularly spaced data. The introduced algorithm is of a forward-backward type; the output of this algorithm can be used for smoothing. This brings inference in the model class in line with other similar models.


**Questions:**

Could you give one example of an application of such a model? Where would you find this kind of data? The method is presented in entirety without any reference to applications. It would be good to have an idea of the practical scale of applicability of this method.

From a numerics point of view, I don't really understand the discussion in the first paragraph of section 4.2, which seems somewhat crucial to the practical implementation of the method. You say that using (13) is numerically more costly but at the same time more stable. This is in comparison to the method presented in appendix C.

Minor comments:

typo on line 333 "where" should be "were"
typo on line 347 "testet" should be "tested"


**Limitations:**

Yes.

**Strengths And Weaknesses:**

Overall, this paper makes a good contribution. I have enjoyed reading the paper and it is well-written and (together with the supplementary material) provides sufficient detail to understand the proposed methodology. The derivation of the forward-backward equations for the considered model is non-trivial and at the same time a very natural question to ask. A more principled inference method as an alternative to existing ad hoc ones is certainly welcome.

Why not provide a description of the (main) forward-backward algorithm step by step in the main paper instead of leaving in to supplementary section C, where it is nicely described?

---

> ### Author Response · Authors · 2022-08-02
> **Initial response / algorithm in main text / applications / choice of smoothing equation**
>
> We thank the reviewer for their careful review of our work. We tried to correct mentioned typos and grammatical errors and to conform to minor comments in a minor revision of the document issued for this rebuttal. In this revision, we also reworked our figures to include proper legends and axis descriptions. In the following we want to respond to major comments and questions which came up during review.
>
> As a result of content selection for the main text, we decided to leave the step-by-step description of the forward-backward and Viterbi algorithms to the supplementary material as referenced in the dedicated sections. We understand that such a description as part of the main text also helps to put the content into place. In a next major revision of the main text, we will consider an inclusion of the algorithmic descriptions.
>
> The domain of application for the model mainly covers the one for classical discrete-time HSMM's (c.f. Yu's book from 2016, p. 163-164) and - more recently available - for CT-HMM's (e.g. c.f. Liu 2015). This covers disease progression analysis, human activity recognition, EEG/ECG data analysis, music modelling, to name a few.
>
> In problems tackled with HSMMs, discrete event data is often adjusted to a time-grid or has a high dynamic range with respect to arrival of events (c.f. Cuvillier 2014, Gong 2015). If continuous-time data is available, it is sampled with a fixed frequency to generate the discrete-time observations for the HSMM while it might be useful to sample adaptively to minimize the linear interpolation error, possibly reducing the amount of samples in total as well (c.f. Borst 2015). Tackling these problems efficiently is available when using our adaptive HSMM approach. Combined with e.g. linear interpolation of waiting time densities it keeps computational effort comparable to a classical HSMM and could be able to reduce it if a reduction of the number of steps is possible. Problems tackled for CT-HMM's suffer from the same limitation which lead to the introduction of HSMMs as an extension to HMMs: a fixed waiting time distribution. Exponential waiting times cannot model a specific variance or higher moments suggested by the evidence. This is especially problematic if single states need to exclude extremely short residence times.
>
> In section 4.2 we mention our preference of the ``traditional'' method to compute the Bayesian posterior. The method is traditional in the sense that its discrete-time analogue is the classical method to obtain the Bayesian posterior in HSMM's (c.f. Yu's book from 2016). The availability of an integral formulation while at the same time knowing the integrand across the whole domain generally favors interpolation techniques compared to extrapolation techniques. Small errors in the computed currents might average out in more cases using an integral formulation. Having said that, we can understand the irritation, since this largely depends on the ODE-solving/quadrature method used. In our minor revision of the main document issued for this rebuttal, we therefor reformulated the sentence to appear more cautious.

---

### Meta-Review · Area_Chair_PkdK · 2022-08-23

**Recommendation:** Accept
**Confidence:** Certain

**Metareview:**

All of the authors agree that the work meets the NeurIPS standards, with the two lowest-scoring reviewers upping their recommendation from 4 to 5 on rebuttal. The work is described as "a fundamental and important problem" and "timely", "a good contribution".

Reviewer 6Ntb summarises the technical contribution:
"The paper generalizes latent state inference from HSMMs to continuous-time chains in latent space. In this case, the posterior is not simply proportional to the usual forward and backward probabilities. Instead, the transition random variables ("currents") are Markov. The authors take the limit of step size approaching 0 for these currents"
It's clear to me that the the work has been communicated really well, since all of the reviewers were able to grasp the paper and there was very little misunderstanding in the discussions.

There were some recommendations from the reviewers - please ensure these are fixed in the camera-ready version.


**Award:**

No

---

### Decision · Program_Chairs · 2022-09-14

Accept